# A cross-scale framework for evaluating flexibility values of battery and fuel cell electric vehicles

Ruixue Liu [1,11], Guannan He [2,3,4,5,11] ✉, Xizhe Wang[1], Dharik Mallapragada [6], Hongbo Zhao[7], Yang Shao-Horn [6,8,9,10] ✉ & Benben Jiang [1] ✉

Flexibility has become increasingly important considering the intermittency of variable renewable energy in low-carbon energy systems. Electrified transportation exhibits great potential to provide flexibility. This article analyzed and compared the flexibility values of battery electric vehicles and fuel cell electric vehicles for planning and operating interdependent electricity and hydrogen supply chains while considering battery degradation costs. A cross-scale framework involving both macro-level and micro-level models was proposed to compute the profits of flexible EV refueling/charging with battery degradation considered. Here we show that the flexibility reduction after considering battery degradation is quantified by at least 4.7% of the minimum system cost and enlarged under fast charging and low-temperature scenarios. Our findings imply that energy policies and relevant management technologies are crucial to shaping the comparative flexibility advantage of the two transportation electrification pathways. The proposed cross-scale methodology has broad implications for the assessment of emerging energy technologies with complex dynamics.

Deep decarbonization entails high penetration of variable renewable energy (VRE)[1,2] and energy demand electrification[3,4]. The intermittency and uncertainty in VRE generation can pose significant challenges to the energy system's supply-demand balance and reliability[5,6]. Consequently, to address the challenges of VRE generation uncertainty and intermittency, flexibility, which is defined as the ability of the power system to balance energy supply with demand in the VRE system, has become more valuable in energy systems[7]. Many solutions to improving the operational flexibility of energy systems have been proposed, e.g., energy storage[8,9], demand-side electrification, and demand-side response[10]. Transportation electrification, which is essential to deep decarbonization[11,12], can also produce a distributed source of flexibility for VRE integration[13]. In addition to satisfying transportation demands, electric vehicles (EVs) can also function as portable energy storage systems with controllable charging and discharging[14]. Smart charging, which optimizes EV charging schedules based on energy prices and energy system balances, allows EVs to provide demand-side flexibility for VRE integration[15,16]. As the share of

[1]Department of Automation, Beijing National Research Center for Information Science and Technology, Tsinghua University, Beijing, China. [2]Department of Industrial Engineering and Management, College of Engineering, Peking University, Beijing, China. [3]National Engineering Laboratory for Big Data Analysis and Applications, Peking University, Beijing, China. [4]Institute of Carbon Neutrality, Peking University, Beijing, China. [5]Peking University Changsha Institute for Computing and Digital Economy, Beijing, China. [6]MIT Energy Initiative, Massachusetts Institute of Technology, 77 Massachusetts Avenue, Cambridge, MA, USA. [7]Department of Chemical and Biological Engineering, Princeton University, Princeton, NJ, USA. [8]Department of Mechanical Engineering, Massachusetts Institute of Technology, 77 Massachusetts Avenue, Cambridge, MA, USA. [9]Research Lab of Electronics, Massachusetts Institute of Technology, 77 Massachusetts Avenue, Cambridge, MA, USA. [10]Department of Materials Science and Engineering, Massachusetts Institute of Technology, 77 Massachusetts Avenue, Cambridge, MA, USA. [11]These authors contributed equally: Ruixue Liu, Guannan He. ✉e-mail: gnhe@pku.edu.cn; shaohorn@mit.edu; bbjiang@tsinghua.edu.cn

EVs increases, the flexibility value will become increasingly significant and play a greater role in evaluating decarbonization pathways of energy and transportation systems[13,17].

Regarding transportation electrification, the two main options are battery EVs (BEVs) and hydrogen fuel cell EVs (FCEVs), with BEVs dominating light-duty fleets[14] and FCEVs exhibiting promising heavy-duty applications[18]. Compared to BEVs, FCEVs have longer driving ranges (over 500 km), as well as faster and more convenient refueling (a few minutes, similar to conventional vehicles)[19,20]. However, challenges including higher costs, lack of hydrogen refueling infrastructure, and limited longevity of fuel cells must be overcome for larger-scale FCEV development[18,21]. Thus, the efficiency, value and development of FCEVs in the short term remain uncertain[22,23]. In the future, the increasing scale of the hydrogen economy might decrease the cost of the hydrogen supply, improving the advantages of FCEVs[24]. In addition to FCEVs, researchers have also explored heavy-duty battery-electric trucks[25–27]. Regarding BEVs in heavy-duty applications, battery degradation, and charging speed are major barriers[28]. Even though many existing studies have compared the BEVs and FCEVs in a life cycle assessment framework incorporating variability[29–32], the values of flexible charging of BEVs and FCEVs to the flexibility of the whole energy systems are still undiscovered and quantified. As flexibility becomes increasingly important in designing energy systems with high penetration rates of VRE, accurate evaluation of the flexibility values of BEVs and FCEVs can provide valuable insights into decarbonization pathway design in the transport and energy sectors.

Current studies on energy flexibility mainly focus on assessing the need for flexibility in VRE systems[33–35] and modeling and analyzing the flexibility offered by different types of supply-side and demand-side resources[36,37]. The valuation of demand flexibility is usually quantified by the cost reduction or benefits of the energy system owing to demand-side flexibility[38,39]. However, investigations into the flexibility value evaluation based on coupled sectors are limited. To the authors' knowledge, the flexibility values of BEVs and FCEVs have not been compared using a sector-coupling energy system optimization model with a high temporal resolution and a full spectrum of energy supply chain technologies in the power and hydrogen sectors. In addition, most studies do not account for the degradation cost of providing flexibility through switching between various charging protocols of BEVs[40–42]. Although many models have been developed to simulate battery degradation[43–46], the implications of the degradation cost for EV charging flexibility are still unclear. The charging strategy and service temperature significantly affect battery degradation, while flexible fueling of FCEVs does not incur evident further degradation. Therefore, accurate estimation of the battery degradation cost under various charging protocols and temperatures is crucial for flexibility comparison of BEVs and FCEVs[47,48].

In this work, we employ an interdisciplinary approach that synthesizes both macro-level and micro-level models to compare the flexibility values of BEVs and FCEVs by computing the system least-cost reduction resulting from EV flexible charging. For the macro-level cost computation, a sector-coupling energy system optimization model, namely, DOLPHYN (see "Code Availability" for the detail), is adopted that minimizes the total capital cost (CAPEX), operational cost (OPEX), emission cost (involving any taxes imposed), and degradation cost considering electricity and hydrogen production, storage, transmission, and demand. The micro-level model based on porous electrode theory (PET) is for analyzing BEV degradation under various charging strategies and temperatures, with its outputs fed back into the macro-level model. We highlight the value of this cross-scale model framework in analyzing the implications of physical characteristics for energy technology comparison and pathway design. Our results show that the net flexibility value of BEVs is significantly reduced due to inevitable battery degradation. Policies and external factors such as the overall hydrogen demand scale (in sectors such as transport,

industry, and heating), hydrogen pathway, carbon price, EV charging speed, and service temperature affect the comparative advantage of transportation electrification (BEVs or FCEVs) for providing greater flexibility. In contrast to the mixed hydrogen pathway, including both natural gas with carbon capture and storage (NG with CCS) and electrolytic generation, the flexibility of BEVs could be less valuable under the electrolytic hydrogen-only pathway, and FCEVs could thus become better flexibility providers under relatively more scenarios. Moreover, the degradation cost of BEVs evidently affects the flexibility value by decreasing charging duration and temperature, which emphasizes the importance of BEV fast charging protocol optimization and thermal management to improve the battery lifetime for higher BEV flexibility values.

## Results
### Model overview
Since less flexibility in the VRE system will increase the system's lowest cost, the flexibility value was quantified as the reduction in the system minimum cost resulting from EV flexible charging, which was computed as the difference between the system costs with and without considering flexible EV charging using an energy system optimization model (DOLPHYN). The research framework for the flexibility of BEVs and FCEVs is depicted in Fig. 1. The DOLPHYN is a sector-coupling planning and decision optimization model to minimize the cost of the low-carbon power network (see "Methods" section). It optimized the costs to identify the most effective and efficient design and operation of the energy system, by modeling the coupling and conversion of different energy sectors and revealing the competition and complementary among different technologies. With this model, we simultaneously optimized infrastructure investments and operations across both electricity and $H_2$ supply chains incorporating production, storage, transmission, end-use consumption, and carbon emission, to obtain the lowest total system cost to meet the electricity and $H_2$ demands. The system cost consists of the CAPEX, OPEX, emission cost, and BEV degradation cost, which is minimized subject to various technological and system operation constraints enforced over representative periods (weeks in this study) at an hourly resolution as well as policy factors such as the $CO_2$ emission price. The representative weeks were selected based on K-means clustering from 7-year data of renewable generation and electricity demand. Linear programming solved by Gurobi with barrier methods[49] was applied to address the optimization problem under four operational constraints (detailed constraints are available in "Methods" section).

As shown in Fig. 1a, we customized the DOLPHYN model to optimize the energy systems without and with flexible charging, taking the optimized cost of the former one as the benchmark cost. The flexibility value of EVs was obtained by the difference value between the benchmark cost and the system least-cost with EV flexible charging. With regard to BEVs, the DOLPHYN model was also customized to the energy system involving battery degradation cost. Then the flexibility value of BEVs considering battery degradation was computed by subtracting the system least cost with flexible charging and degradation from the benchmark cost. More details are available in "Methods" section and Supplementary Fig. 1.

We modeled the EV flexible charging as deferrable demand, which is the flexible consumption of hydrogen or electric power for EVs. It implies that the EVs do not need to be charged or refueled immediately after their arrivals. Our study on flexibility assessment of EVs was based on a marginal perspective, which means that the flexible charging demand is less than the total EV charging demand at times when flexible charging demand is needed. Thus, the scale of deferrable EV demand was kept within a relatively small range in our analysis. We took this assumption because the future EV charging distribution and the ratio of deferable demand are highly uncertain and may deviate from the current patterns.

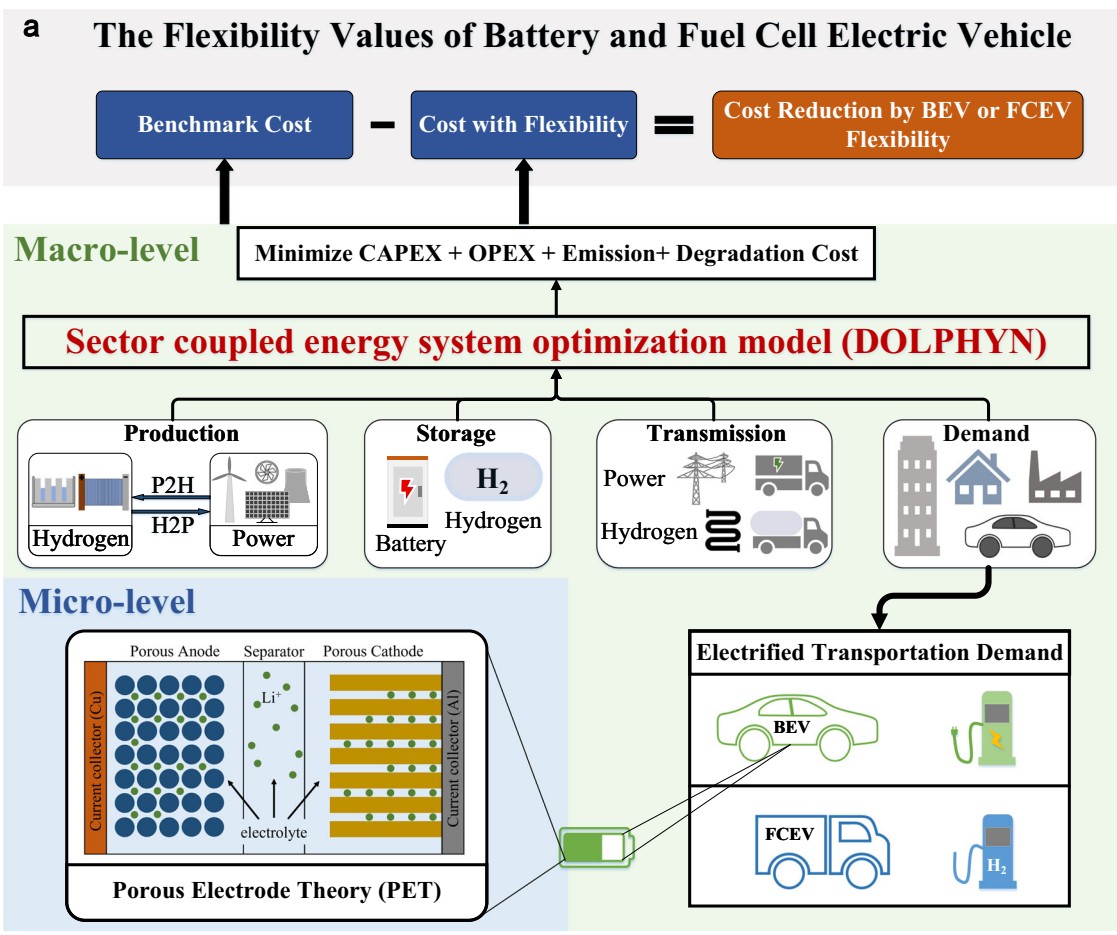

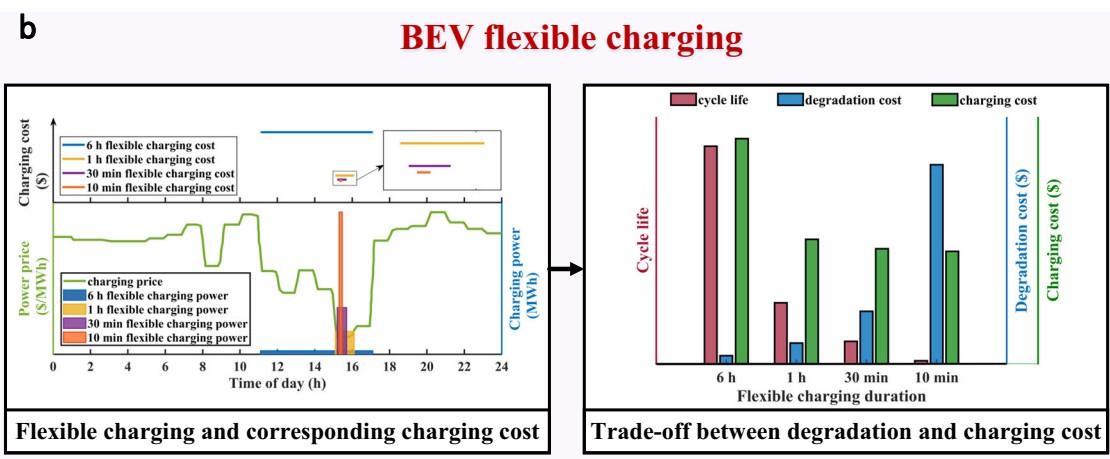

**Fig. 1 | Schematic of the evaluation of the flexibility values of battery electric vehicles (BEVs) and fuel cell electric vehicles (FCEVs). a** The flexibility value was computed as the system's least-cost reduction due to BEV or FCEV flexibility. Both macro-level model and micro-level analysis were synthesized to compute flexibility values. At the macro level, the sector-coupling DOLPHYN model was adopted to optimize the low-carbon energy system for obtaining the minimum system cost, considering the capital cost (CAPEX), operational cost (OPEX), emission cost (EMISSION), and BEV degradation cost. To involve the BEV degradation cost, a micro-level PET-based model was applied to simulate battery degradation and cycle life under various charging durations, which is regarded as one of the inputs of the DOLPHYN model. **b** BEV flexible charging. The flexibility was embodied in that BEVs could be charged under high VRE generation and a low power price. When the period with the highest VRE and the lowest power price is short, faster charging for a shorter charging duration is preferred to lower the charging cost by completing charging at the lowest power price. Nevertheless, a shorter charging duration with a higher charging power could accelerate battery degradation, leading to extra degradation costs and reduced flexibility value. These trade-offs are considered when evaluating the flexibility value of BEVs in this work.

The flexible charging of BEVs involves a trade-off between the charging speed and degradation costs (Fig. 1b). During flexible BEV charging, system costs can be reduced by charging when VRE is abundant and the power price is low. Since the power price varies as VRE generation fluctuates, faster charging can facilitate a lower average charging price within a certain time window, as shown in the left panel of Fig. 1b. From this perspective, fast charging could reduce charging/system costs. Whereas, faster charging typically results in accelerated degradation, a shorter cycle life, and thus a higher degradation cost (see the right panel of Fig. 1b), which might pose challenges to providing greater flexibility value in the battery's subsequent service life.

To incorporate the BEV degradation cost into the system cost, battery degradation under various BEV charging durations and temperatures must be analyzed before applying the DOLPHYN model. It is assumed that the chargers are enough and available for BEV charging preconditioned on modeling EV flexible charging as deferrable demand. The corresponding degradation cost was computed by dividing the cost of the battery replacement by its cycle life (see "Methods" section). A PET-based model (see "Methods" section) was adopted to simulate battery degradation in response to various flexible charging demands and environmental temperatures. According to our PET-based model, the lifetime of batteries is nearly 3000 cycles when charged for 60 min at 25 °C, which is regarded as the benchmark in our work. The cycle life is reduced with decreasing charging duration and temperature (see "Degradation of BEVs with flexible charging" section). The battery cycle life obtained via the PET-based model and relevant parameters (see Supplementary Tables 5, 6) were used as the inputs of the DOLPHYN model to compute the BEV degradation cost (see "Methods" section). After including the degradation cost in the system cost, the system cost reduction due to BEV flexibility could then be obtained.

The mechanisms to utilize the flexibility of FCEVs are different from those of BEVs. Since the hydrogen supply chain including the storage process is closely related to FCEVs flexibility (i.e., the flexibility value of FCEVs in this study is indeed attached to the flexibility of the hydrogen supply chain, specifically the storage capacities at the hydrogen refueling stations and transmission between zones), we modeled the refueling of FCEVs as a dispatchable demand. Specifically, hydrogen production, storage, compression, and transmission are included in the hydrogen supply chain, as shown in Fig. 1 and supplementary Fig. 2. Among the four elements, hydrogen storage, and transmission are two direct measures to provide the required flexibility for the supply chain. Regarding hydrogen storage, in addition to stationary storage, mobile storage including trucks and pipelines was also considered in our model, enabling hydrogen shifting in space and time while being shared across the whole hydrogen network to match hydrogen demand. In other words, trucks and pipelines were mainly modeled as both transmission and storage functions to provide flexibility to the hydrogen supply chain. The optimization model incorporates the hydrogen flexible storage and transmission scheduling. It means that the decision variables of the DOLPHYN model involve the capacities of hydrogen storage and transmission between zones. These variables were optimized by the model in response to the hydrogen flexible demand. The H$_2$ demand for each zone was developed based on available fuel consumption data and hourly charging profiles for mainly heavy-duty FCEVs and the relative penetration of FCEVs. Evidently, flexibility is preconditioned on the deployment of the FCEV fleet, and therefore, we attributed this flexibility to the FCEV pathway.

Flexible fueling of FCEVs does not incur extra degradation, with the normal fueling time of hydrogen storage short enough for the time windows considered in this study. Regarding electrolyzers in the DOLPHYN model, their power inputs were assumed to be limited under the rated power capacities, so no overloading was allowed.

Under this operational constraint, producing hydrogen with electrolysis in a shorter window, i.e., increasing the electrolyzer loading from 50% to 100%, does not result in notable extra degradation and could even limit degradation under some scenarios[50], i.e., increasing from 10% to 60%. The fast-charging setting investigated in this work is unlikely to accelerate electrolysis degradation. Thus, extra degradation of FCEV flexible charging was not considered.

## Scenario setup
We used a case study of the U.S. Northeast region to implement the optimization model and compare the flexibility values of BEVs and FCEVs (mainly for heavy-duty EVs), under a variety of demand, technology, and CO$_2$ price scenarios for 2050. In view of different charging requirements and demands, we classified five scenarios for flexibility value evaluation based on variable control. The flexibility values of BEVs and FCEVs were evaluated and compared under scenarios with various charging modes, battery replacement costs, hydrogen pathways, and service temperatures, considering different carbon prices, overall hydrogen demand scales, and flexible demand scales (signified as deferrable demand in this work). These scenario parameters reflect the decarbonization progress, policies, and technology development. For instance, the deeper decarbonization process might show higher hydrogen demand, larger EV flexible charging demands, and higher carbon prices. To model flexible EV charging, we considered two load-shifting settings (i.e., fast charging and normal charging) at room temperature with the electrolytic-only hydrogen pathway applied. The time window of the fast-charging setting is 1 h, while that of the normal charging setting is 6 h. Different battery replacement costs from \$50/kWh to \$200/kWh were considered in the normal charging case. For the fast-charging scenario, an average charging case and an extreme fast-charging case were further discussed. When comparing the flexibility values of BEVs and FCEVs under different hydrogen pathways or temperatures, the BEV average fast-charging setting with a \$100/kWh battery replacement cost was applied. Two hydrogen pathways include the electrolytic-only hydrogen and the mixed hydrogen pathway. Temperatures considered in this work were 25 °C, 10 °C, and 0 °C, respectively, according to the annual climate of the U.S. Northeast region. Lower temperatures are discussed because studies have illustrated that lower temperatures evidently affect battery charging. All scenarios involved in this work are summarized in Table 1.

## The significance of considering BEV degradation
The net values of flexible FCEV and BEV charging with and without battery degradation involved were compared in the form of the system cost reduction under both the fast (Fig. 2) and normal (Fig. 3) charging settings, with a greater cost reduction representing a larger flexibility value. Regarding the fast-charging setting, we assumed that both BEVs and FCEVs require a maximal charging time of 1 h and that the charging load may be shifted to any time within the 1-h window. The fast-charging setting characterizes the charging of long-haul EVs, especially heavy-duty EVs, which are sensitive to the charging time[51]. In contrast, the normal charging setting assumes a 6-h charging window and refers to the charging of short-haul EVs.

The net value of flexible BEV charging is much smaller after considering the extra degradation cost of BEV batteries in both cases under a variety of scenarios, at only less than a 2.0% reduction in the total minimal system cost for fast charging (Fig. 2) and 3.1% for normal charging (left panel in Fig. 3a) with the same battery replacement cost of \$100/kWh, relative to the maximal value of 7.8% when degradation is not considered. Because of battery degradation, the flexibility values of BEVs decrease by up to 6.9% for the fast-charging setting (left panel in Fig. 2a) and 4.7% for the normal charging setting (left panel in Fig. 3a) at a CO$_2$ price of \$1000/tonne and an overall H$_2$ demand scale of 1 Mtonne/year.

**Table 1 | Summaries of scenarios for comparing flexibility values of BEVs and FCEVs**

| Scenarios | | | Charging duration or Time window | Temperature | Battery replacement cost | Hydrogen pathway |
|---|---|---|---|---|---|---|
| Various charging modes | Fast charging | FCEVs (%) | 1 h | 25 °C | – | Electrolytic-only |
| | | BEV average (%) | 15 min, 30 min, 45 min, 60 min | 25 °C | $100/kWh | |
| | | BEV extreme (%) | 10 min | | | |
| | Normal charging | FCEVs (%) | 6 h | 25 °C | – | |
| | | BEV $50–100 (%) | | 25 °C | $50–100/kWh | |
| | | BEV $100–150 (%) | | | $100–150/kWh | |
| | | BEV $150–200 (%) | | | $150–200/kWh | |
| Hydrogen pathway | Mixed | FCEVs (%) | 1 h | 25 °C | – | Mixed |
| | | BEV average (%) | | 25 °C | $100/kWh | |
| | Electrolytic-only | FCEVs (%) | | 25 °C | – | Electrolytic-only |
| | | BEV average (%) | | 25 °C | $100/kWh | |
| Service temperature | | FCEVs (%) | 1 h | 25 °C | – | Electrolytic-only |
| | | BEV average 25 °C (%) | | 25 °C | $100/kWh | |
| | | BEV average 10 °C (%) | | 10 °C | | |
| | | BEV average 0 °C (%) | | 0 °C | | |

These results indicate the significance of considering battery degradation for both the fast and normal charging settings (see "Degradation of BEVs with flexible charging" section) and the value of integrating a micro-level physical model with a macro-level planning model. On this basis, to ensure greater practicality, all the following analyses of BEV flexibility consider the battery degradation cost. In the future, BEV battery degradation is expected to be improved, profiting from developed battery technologies. By then, in contrast to normal charging, fast charging scenarios with a relatively shorter cycle life and higher degradation cost might be more common[52–54].

## Flexibility values of BEVs and FCEVs under fast and normal charging settings

In regard to the fast-charging setting, we further considered two cases—an average case termed "BEV Average" and an extreme case termed "BEV Extreme"—with the same battery replacement cost of $100/kWh. In both cases, we adopted the degradation in BEV charging within 60 min as the benchmark. The cost reduction values of the four cases (FCEV, BEV without considering degradation, BEV Average and BEV Extreme) are compared under the different scenarios in Fig. 2, with different overall $H_2$ demand scales in Fig. 2a–c and various carbon prices of $1000, $100, and $0/tonne from left to right. In the "BEV Average" case, we assumed that BEV arrivals are uniformly distributed, and we used the average of the extra degradation costs associated with charging times of 15, 30, 45, and 60 min over the 60-min benchmark (see "Methods" section). The average cycle life is approximately 1600 cycles, which is a reduction of approximately 45.5% from the benchmark level.

After considering BEV degradation with BEV Average fast-charging assumptions, the flexibility values of BEVs are higher than those of FCEVs under most scenarios. The flexibility values of FCEVs outperform those of BEVs only under the scenarios with a higher $H_2$ demand of 4 Mtonne/year and a low/medium carbon price of $100 or $0/tonne (middle and right of Fig. 2c). Relatively less flexibility is required when the carbon price is lower, owing to VRE discouraging. Under this circumstance, a higher hydrogen demand implying the strong coupling of power and hydrogen sectors makes FCEV

predominate over BEVs as the flexibility provider. Under the remaining scenarios, FCEVs tend to exhibit more potential for flexibility with increasing deferrable demand (representing the rate of EVs flexible charging). In other words, when the decarbonization level increases with a higher overall $H_2$ demand scale of 2 or 4 Mtonne/year, FCEVs are more likely to provide systems with greater flexibility values. In the future, if the battery lifetime performance is improved by twofold and the battery replacement cost is reduced by half, the resulting battery degradation cost savings could improve the flexibility values of BEVs considering fast charging settings by up to 4% under some scenarios with a lower overall $H_2$ demand scale and higher carbon prices.

In the BEV Extreme fast charging case, we assumed that BEVs were charged in 10 min and used the extra degradation cost over the benchmark level (60 min of charging). Clearly, as shown in Fig. 2, when BEVs are charged extremely fast throughout the whole service life (the BEV Extreme case), their corresponding net flexibility values are less than those in the BEV Average case by approximately 0.8%, owing to the accelerated degradation with an approximately 91.1% reduction in the battery lifetime (see "Degradation of BEVs with flexible charging" section). Under this condition, the flexibility values of BEVs are less than those of FCEVs on more occasions with a maximum gap of approximately 0.4%. These results illustrate the remarkable influences of fast charging on the resulting BEV flexibility values and the importance of battery charging protocol optimization and degradation mitigation to the selection of the transportation electrification pathway.

Regarding the normal charging setting, we considered three battery replacement cost ranges, as shown in Fig. 3: "BEV $50-$100/kWh" with yellow shadow, "BEV $100-$150/kWh" with a purple shadow, and "BEV $150-$200/kWh" with a green shadow. When increasing the BEV and FCEV charging-time windows from 1 h to 6 h given a $100/kWh battery replacement cost, the cost reduction of BEVs exceeds that of FCEVs under scenarios with a higher carbon price and a lower overall $H_2$ demand scale (Fig. 3a, left and middle of Fig. 3b, and left of Fig. 3c), by a maximum of 2.0% given a carbon price of $1000/tonne and an $H_2$ demand of 1 Mtonne/year (left of Fig. 3a). In contrast to the 1 h conditions depicted in Fig. 2, when the charging time window is increased to 6 h, the maximum increment in the BEV flexibility value under the

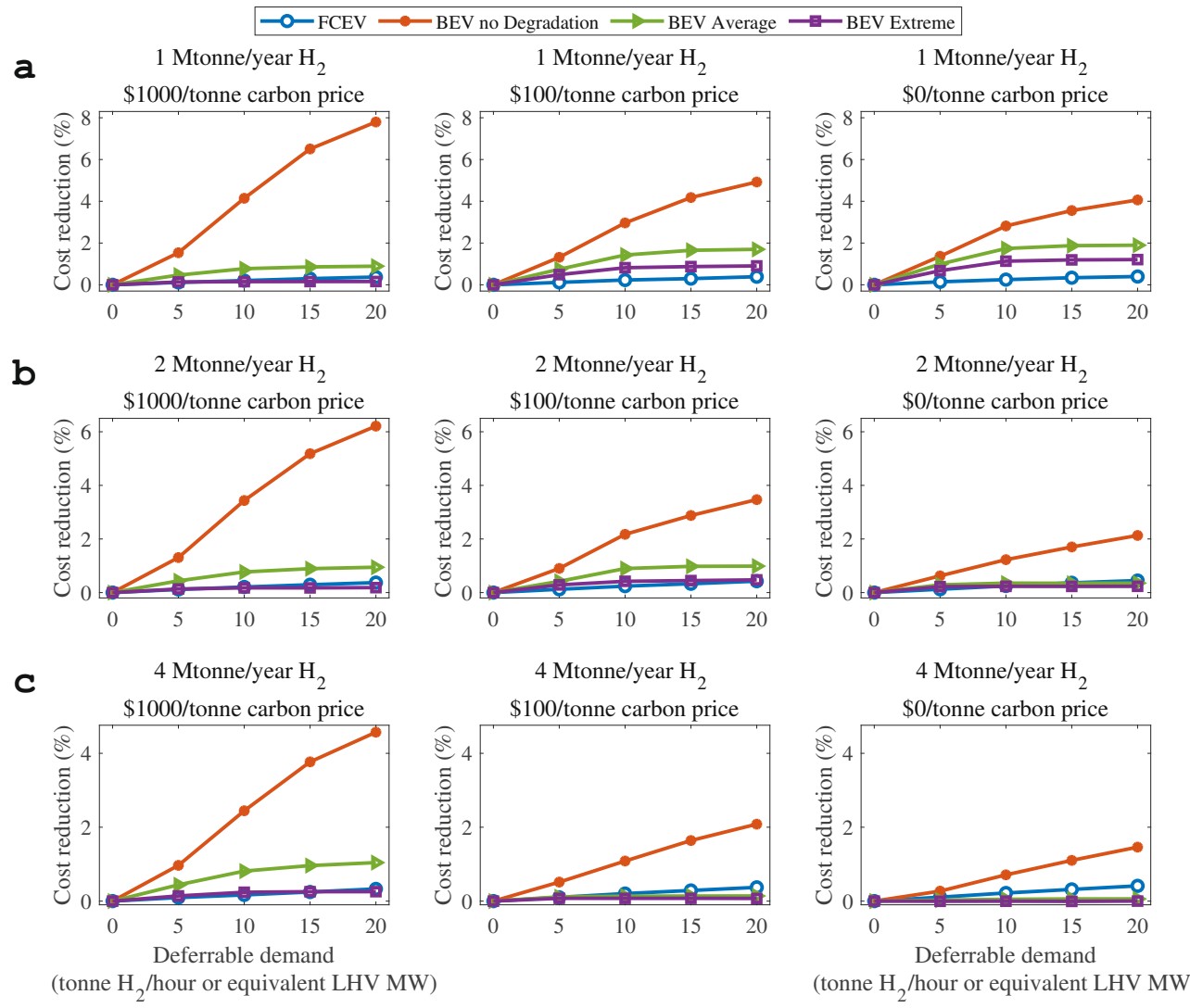

**Fig. 2 | Net values of flexible battery electric vehicle (BEV) and fuel cell electric vehicle (FCEV) fast charging with a charging window of 1 h under scenarios with different carbon prices, H₂ demands, and flexible electric vehicle (EV) capacities. a** The overall H₂ demand scale is 1 Mtonne/year. **b** The overall H₂ demand scale is 2 Mtonne/year. **c** The overall H₂ demand scale is 4 Mtonne/year. The panels from left to right in each subfigure show carbon prices of $1000/tonne, $100/tonne, and $0/tonne. The blue lines with circle markers denote the cost reduction due to the flexibility of FCEVs. The orange, green, and purple lines indicate the flexibility values of BEVs without BEV degradation (marked as "BEV no

Degradation"), with BEV degradation combining 4 charging-time protocols (Supplementary Fig. 8) in the same proportion (marked as "BEV Average"), and with BEV degradation under extremely fast charging in only 10 min (marked as "BEV Extreme"), respectively. Deferrable demand here is the flexible consumption capacity of hydrogen or electric power for EVs, measured by the maximum deferred capacity per zone in tonne/hour for FCEV and MW for EV. A larger deferrable demand means a larger level of electric vehicle flexible charging. LHV is the abbreviation of lower heating value.

various scenarios reaches approximately 2% with a battery replacement cost of $100/kWh. This is mainly caused by a longer battery lifetime and less degradation owing to a relatively slower charging speed. In addition, when the battery replacement cost is reduced from $200/kWh to $50/kWh, the flexibility values of BEVs increase at increasingly higher rates. For instance, as shown on the left panel of Fig. 3a, when the deferrable EV demand is 20 tonne H₂/hour (~66 MW), the flexibility value of BEVs increases from 1.9% ($200/kWh) to 2.4% ($150/kWh), 3.1% ($100/kWh), and 4.4% ($50/kWh). This growing effect occurs because the reduced battery degradation cost not only lowers the system cost but also encourages the deployment of flexible BEV charging.

**Flexibility values under different hydrogen pathways**
Whether the hydrogen pathway of NG with CCS, which produces hydrogen based on fossil fuel with CCS, should be considered has been debated because of the potential upstream carbon emissions in fossil fuel production[55,56]. Here, we compute and evaluate how the flexibility

values of BEVs and FCEVs change as we switch from the mixed hydrogen pathway, including both NG with CCS and electrolytic generation, to electrolytic hydrogen only. Figure 4 shows the impact of a deferrable EV demand of 15 tonne H₂/hour on the total system cost and the VRE (including wind and solar), combined-cycle gas turbine (CCGT) with and without CCS, hydrogen storage and battery storage capacities, with the blue bars indicating the mixed hydrogen pathway with NG with CCS and the green bars indicating the electrolytic hydrogen only pathway without NG with CCS. Negative values denote decreases in corresponding capacity resulting from EV flexible charging, and vice versa. For FCEVs, the optimized hydrogen storage (consisting of both stationary and mobile storage) is reduced as increasing of the deferrable demand. More reduced hydrogen storage is observed under scenarios with higher H₂ demands. This can be explained that with more flexibility provided by the FCEV fleet, less flexibility is required for the storage process owing to the relatively higher cost to achieve flexibility than FCEVs. The results illustrate how

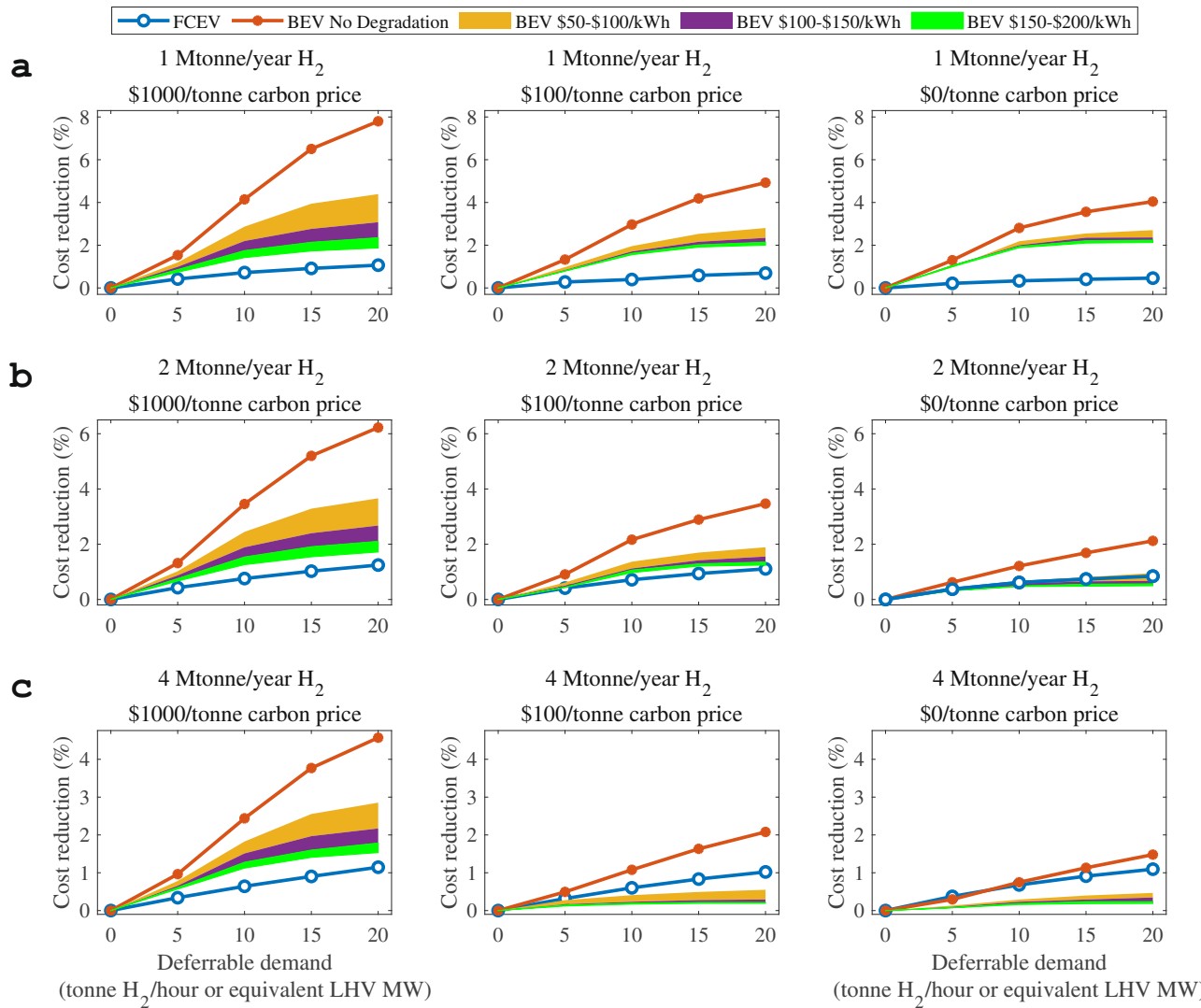

**Fig. 3 | Net values of flexible battery electric vehicle (BEV) and fuel cell electric vehicle (FCEV) normal charging with a charging window of 6 h under scenarios with different carbon prices, H₂ demands, and flexible electric vehicle (EV) capacities. a** The H₂ demand is 1 Mtonne/year. **b** The H₂ demand is 2 Mtonne/year. **c** The H₂ demand is 4 Mtonne/year. The panels from left to right in each subfigure show carbon prices of $1000/tonne, $100/tonne, and $0/tonne. The blue line with white circles denotes the cost reduction due to the flexibility of FCEVs, while the orange line denotes that of BEVs without considering degradation. The system cost reduction of BEVs considering degradation given a $50–$100/kWh battery replacement cost is marked as "BEV $50-$100/kWh", with a yellow shadow, while the scenarios considering $100-$150/kWh and $150-$200/kWh battery replacement costs are marked as "BEV $100-$150/kWh" with a purple shadow and "BEV $150-$200/kWh" with a green shadow, respectively. LHV is the abbreviation of lower heating value.

the FCEV flexibility value interacts with the hydrogen storage process and is attached to the flexibility of the supply chain.

The results show that ruling out hydrogen production from NG with CCS could more notably influence the flexibility value from BEVs than that from FCEVs (Fig. 4 and Supplementary Figs. 4 and 5, respectively). The system cost reductions resulting from FCEV flexible charging under both hydrogen pathways are similar, while those of BEVs under the electrolytic hydrogen-only pathway are remarkably less, suggesting reduced flexibility values of BEVs, especially when the H₂ demand is higher. The high sensitivity of the BEV flexibility value to the hydrogen pathway can be explained by the stronger coupling of the hydrogen supply chain to the power system, through electrolysis, and the substitution effect of the flexible hydrogen supply chain to BEVs. When the electrolytic hydrogen-only pathway is adopted, the hydrogen supply chain becomes more greatly coupled with the power system through electrolysis and provides higher flexibility. In the coupled system, less expensive hydrogen storage could lower the cost of electricity storage and encourage VRE use, substituting the role of flexible BEV charging.

## Effects of the service temperature on EV flexibility values

In practical cases, FCEVs and BEVs are used at various environmental temperatures. For example, in our case study focused on the U.S. Northeast region, where the annual average temperature ranges between 8–15 °C, the local temperature hovers around 25 °C for approximately a quarter of the year. For more than half the year, it remains near 10 °C, and occasionally is below 0 °C during winter. Based on the BEV Average case and the electrolytic-only hydrogen pathway, the flexibility values of BEVs and FCEVs under different temperatures (i.e., 25 °C, 10 °C, and 0 °C) were analyzed considering BEV battery degradation.

The flexibility values of BEVs significantly decrease with decreasing temperature under all carbon prices and H₂ demand scenarios (Fig. 5). The reduced flexibility value could mainly be attributed to the higher cost induced by faster battery degradation at a lower temperature. The battery simulation results show that the battery lifetime using 60-min charging is reduced from approximately 3000 cycles to approximately 1400 cycles when lowering the temperature from 25 °C

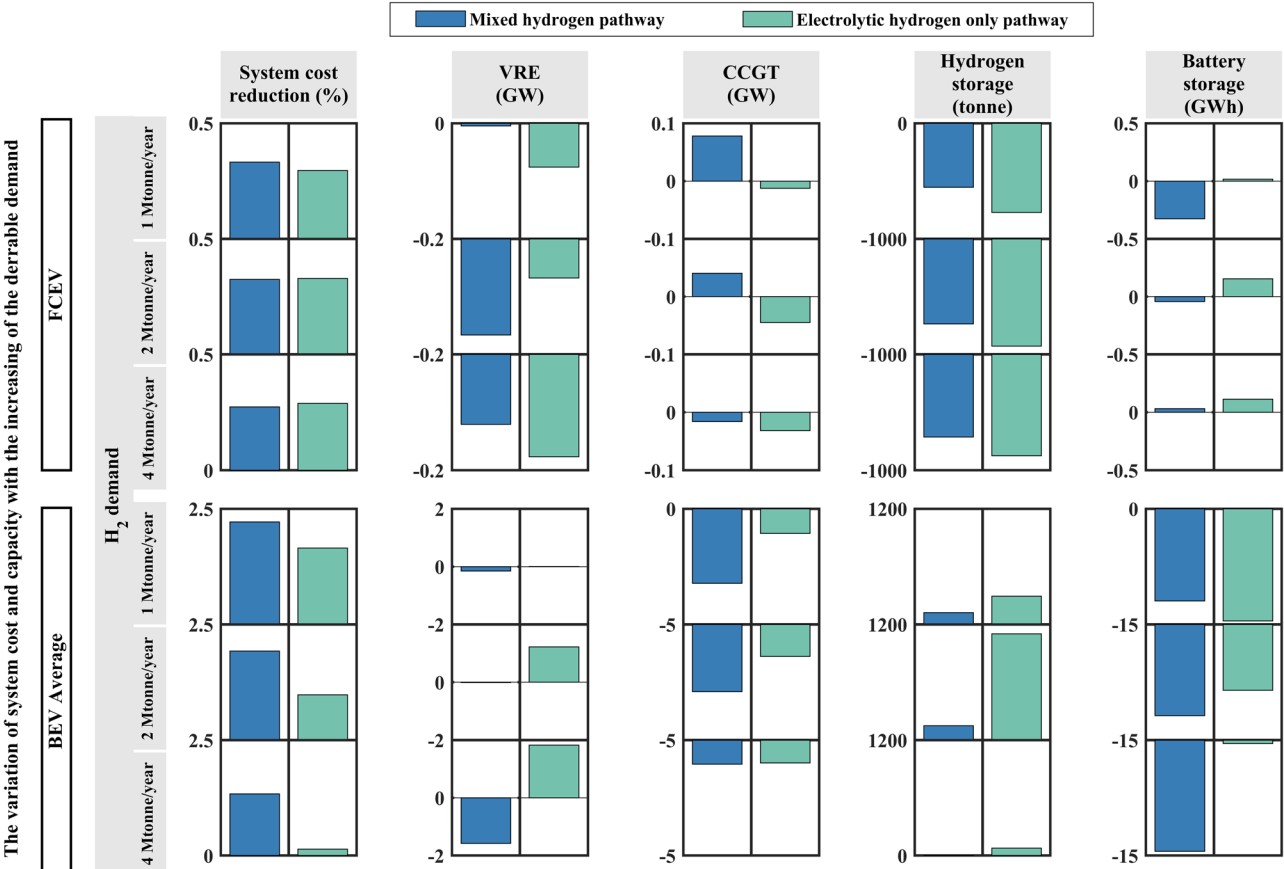

**Fig. 4 | Impact of the deferrable demand on the variation of system cost reduction, variable renewable energy (VRE) capacity, combined-cycle gas turbine (CCGT) capacity, hydrogen storage capacity and battery storage capacity when the carbon price is $100/tonne.** The blue bars and green bars denote the mixed hydrogen pathway (using both electrolysis and steam methane reformer (SMR) technology with carbon capture and storage (CCS)) and the electrolytic-only pathway, respectively. The panels from left to right show changes in the system cost reduction, VRE capacity, CCGT capacity, hydrogen storage capacity, and battery storage capacity, in sequence. The bars from top to bottom denote the $H_2$ demand of 1 Mtonne/year, 2 Mtonne/year, and 4 Mtonne/year, respectively. BEV is short for battery electric vehicle while FCEV is short for fuel cell electric vehicle.

to 10 °C (for more information on battery degradation, see "Degradation of BEVs with flexible charging" section and Supplementary Table 6). As temperatures decrease, the accelerated battery degradation diminishes the benefits of BEVs as flexibility providers. The disparity in BEV flexibility values between 10 °C and 0 °C is more pronounced than that between 25 °C and 10 °C, due to the nonlinear increase in degradation costs as temperatures drop. When the temperature is reduced to 0 °C, FCEVs outperform BEVs in more cases with a deferrable demand over 10 tonne $H_2$/hour due to the particularly rapid battery degradation and the corresponding increased cost at this temperature. The clear reduction in BEV flexibility values with decreasing temperatures indicates that for practical applications aiming to enhance BEV flexibility, effective thermal management should be adopted to prevent BEV batteries from operating and charging at low temperatures.

### Degradation of BEVs with flexible charging
As illustrated in Fig. 1b, a trade-off between the BEV charging cost and degradation cost exists. In addition, there is also contradiction between system benefits and personal profits. In other words, fast charging might be profitable from a system perspective, whereas it is not beneficial to BEV owners. The costs of BEV flexible charging and battery degradation are examined below.

With an increasing VRE penetration rate, various BEV charging protocols are often implemented based on the real-time power system load and the requirements of users over a wide range of environmental temperatures. In terms of the charging cost of BEVs and system benefits, faster charging is expected to be beneficial[57]. However, it is generally acknowledged that battery charging strategies and the service temperature impose considerable effects on battery degradation. Fast charging often accelerates the degradation of batteries and shortens their lifetime, resulting in additional costs. Therefore, the trade-off between the charging cost and battery degradation must be addressed when evaluating the flexibility values of BEVs. In this section, we investigate the degradation and cycle life of batteries under different charging protocols applied to lithium-ion batteries at various environmental temperatures, which is simulated via PETLION[58], a Julia implementation of the PET-based model (see "Methods" section).

Capacity degradation trajectories of lithium-ion batteries under various charging protocols (Supplementary Fig. 8 and Supplementary Note 1) and service temperatures are shown in Fig. 6. The intersection of a given curve and the X-axis denotes the battery cycle life. The model-simulated battery cycle life at 25 °C is similar to the experimental data retrieved from Severson et al.[59], Attia et al.[60], and other measurement results mentioned in Wen et al.[61]. Hence, the obtained cycle life used for the inputs of the DOLPHYN model could be regarded as dependable. As shown in Fig. 6, the battery cycle life notably decreased with decreasing charging time. The gaps between the two curves at the same temperatures (lines with the same colors) illustrate that the battery cycle life sharply decreased with decreasing charging duration. At 25 °C, the cycle life of cells in the extreme cases is reduced by approximately 90% from average case levels. The reduction ratios at

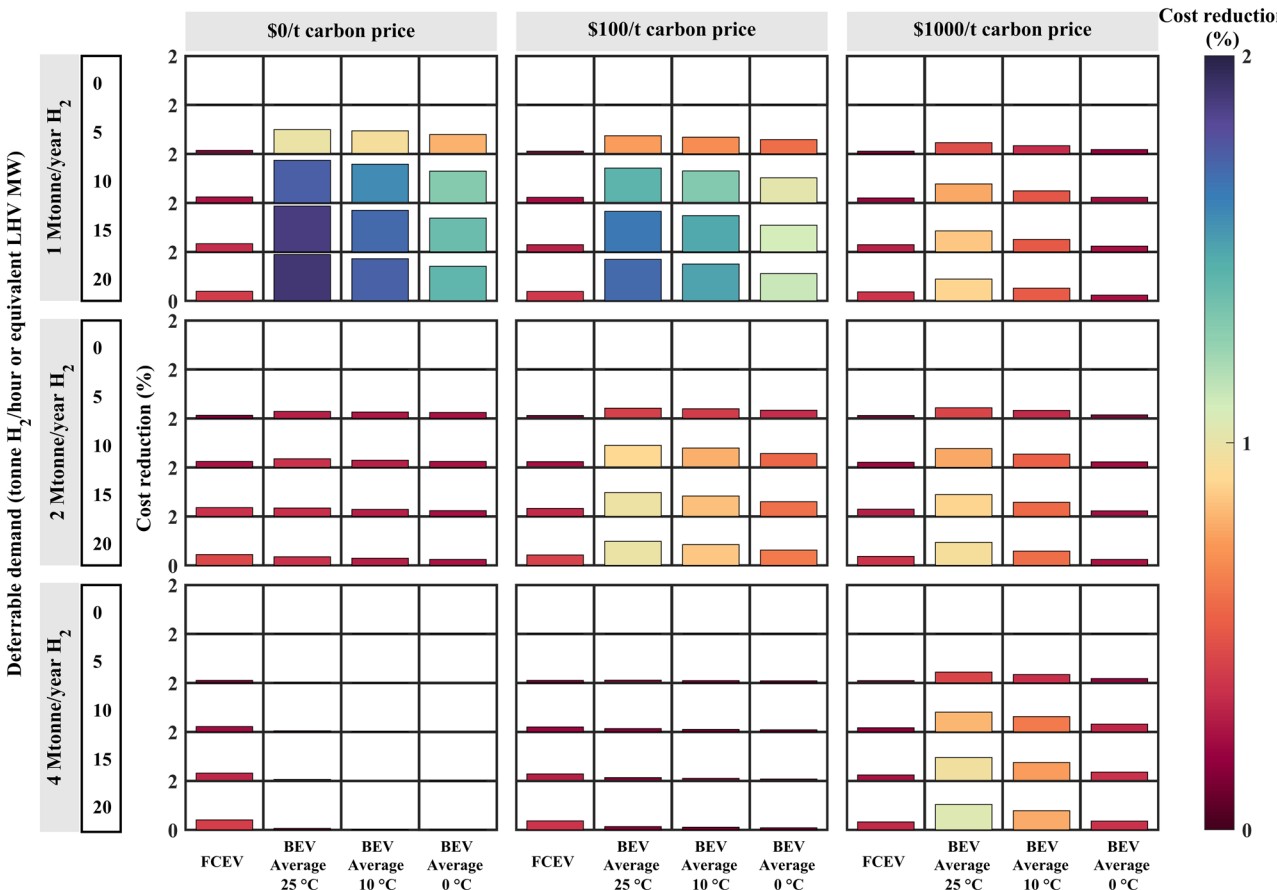

**Fig. 5 | Net values of flexible fuel cell electric vehicle (FCEV) and battery electric vehicle (BEV) fast charging for the different carbon prices and deferrable demands under the various temperatures.** The cost reductions of FCEVs and BEVs in the "BEV Average" case at 25 °C, 10 °C, and 0 °C, are shown by bars of different lengths and colors. The corresponding color legend is provided on the right. LHV is the abbreviation of lower heating value.

10 °C and 0 °C are approximately 93% and 99%, respectively. In summary, batteries dramatically degrade with increasing charging speed (see Supplementary Note 2 and Supplementary Fig. 9 for detailed analyses). This finding explains the notable reduction in the flexibility value after considering battery degradation in the "BEV Extreme" case (Fig. 2).

The results show that extremely fast charging might not be beneficial for improving system flexibility and reducing cost after considering the battery degradation of BEVs. Even though the reduced charging time seems to reduce the charging cost and make BEVs more operationally flexible, the significantly elevated degradation cost may offset the provided advantages or may even surpass them. As a result, BEVs may even attain a smaller flexibility value under the scenario of extremely fast charging (Fig. 2). Therefore, the importance of battery degradation and the corresponding cost should be considered when evaluating the flexibility value of BEVs. In addition, charging optimization that balances the charging cost (or charging speed) and battery degradation is required to achieve higher flexibility values.

By comparing battery degradation using the same charging protocol (same line style) at different temperatures (Fig. 6), it can be observed that a lower temperature is adverse to battery capacity retention, especially in the extreme case (solid lines in Fig. 6). In this case, the battery cycle life at 10 °C is 52.4% smaller than that at 25 °C, while the reduction at 0 °C is 86.8% of that at 10 °C. In the average cases, the battery cycle lives at 10 °C and 0 °C are approximately 62.5% and 44.1% of those at 25 °C. Particularly, in contrast to the average case at 25 °C, the battery lifetime reduction in the extreme case at 0 °C reaches 99.4%. This accounts for the remarkable decrease in the BEV flexibility value at a lower temperature (Fig. 5).

## Discussion

In this work, we evaluated and compared the values of BEV and FCEV flexible charging in terms of system cost reduction. A cross-scale methodological framework integrating macro-level and micro-level models was developed to compute the system cost, considering the battery degradation cost. At the macro level, the DOLPHYN model, which couples various sectors, was applied to compute the minimum system cost, after optimizing decision variables related to power and hydrogen generation, storage capacity, and hydrogen transmission. At the micro level, battery degradation and cycle life under various charging-duration constraints and temperatures were analyzed and calculated through a PET-based model. Then, the degradation results and related cell parameters were used as inputs of the DOLPHYN model to compute the system cost involving BEV degradation.

The results show that the flexibility value of BEVs significantly decreased by more than 4.7% after considering battery degradation. With battery degradation cost involved, whether BEVs or FCEVs is the better choice in view of their flexibility values depends on the decarbonization level, deferrable demand, $H_2$ demand, charging speed, service temperature, and policies such as the hydrogen pathway and carbon price.

Fast charging and lower operating and charging temperatures significantly diminish the flexibility value of BEVs. As a result, under scenarios with extremely short charging durations and low temperatures, FCEVs are more promising flexibility providers, which is primarily due to the substantially reduced battery lifespan in such conditions. Furthermore, escalating battery replacement costs can also reduce BEV flexibility value. Typically, battery replacement costs remain relatively consistent over time. To ensure a prolonged battery

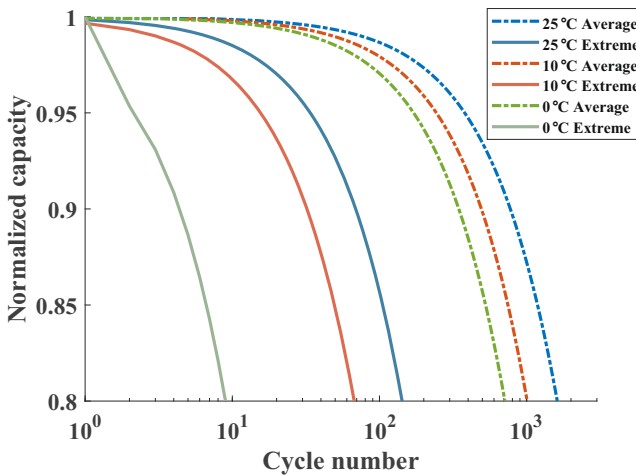

**Fig. 6 | Battery degradation and lifetime under the different temperatures and charging protocols.** "Average" denotes average battery lifetimes using four charging protocols with charging times of 60 min, 45 min, 30 min, and 15 min (see Supplementary Table 6). "Extreme" denotes extremely fast charging with only a 10 min duration. The blue, red, and green lines indicate battery degradation at 25 °C, 10 °C, and 0 °C, respectively. The solid lines denote the extreme cases, while the dotted-dashed lines denote the average cases. When at 25 °C, "Average" is associated with "BEV Average" case in Fig. 2, while "Extreme" is corresponding to "BEV Extreme" case in Fig. 2 (BEV is short for battery electric vehicle). The battery cycle life defined here is the cycle number corresponding to a reduction in the cell capacity to 80% of the nominal capacity[62].

lifespan and greater BEV flexibility, it's advisable to avoid conditions of low temperatures and fast charging. However, this is not always feasible. Thus, advancements in battery technology, particularly concerning charging and thermal performance, are important to bolster BEVs' relative advantages as flexibility provider, since the lifespan and associated degradation costs of BEV batteries are significant when assessing the flexibility value. When the $H_2$ demand is high (2 or 4 Mtonne/year) and the carbon price is low ($100/tonne or $0/tonne), FCEVs show more potential to providing essential flexibility. In most other scenarios, BEVs tend to have a superior flexibility value. Higher carbon prices, which are often associated with medium to deep decarbonization levels, promote a larger penetration of VRE, increased battery storage capacity, and reduced reliance on CCGT for power generation. This surge in flexibility requirements then favors BEVs. Higher $H_2$ demands lead to a tighter integration between the hydrogen and power sectors. Consequently, in such scenarios, the flexibility value of BEVs significantly decreases, making them less competitive compared to FCEVs. Regarding the $H_2$ generation pathway, when the mixed hydrogen pathway is replaced by the electrolytic hydrogen-only pathway, the flexibility value of BEVs is more evidently reduced, on account of the stronger coupling of the hydrogen supply chain to the power system through electrolysis due to higher hydrogen storage and less battery storage. Then, FCEVs could provide a greater flexibility value than BEVs at lower carbon prices and higher $H_2$ demands (Fig. 4).

These are how energy policies greatly influence the comparative advantages of the two transportation electrification pathways in terms of flexibility values. In the current and near terms with relatively less $H_2$ demands, BEVs are expected to still dominate because they offer most of the required flexibility value, while in the long term when the carbon price and hydrogen demand are higher, FCEVs might be more promising even with the battery replacement cost reduced to $50/kWh.

To further increase the flexibility value of BEVs, the battery lifetime should be notably improved in the future. Twofold extended lifetime due to technological improvement may offer more flexibility by an additional 4% system cost reduction. To this end, when BEVs are connected to power grids, smart charging or even vehicle-to-grid

(V2G) systems should be developed to meet system integration and demand by rationally scheduling BEV charging under optimal charging protocols. Within this context, it is essential to achieve a higher flexibility value and lower system cost by adaptively optimizing charging strategies to address the trade-off between the short charging-duration demand and battery degradation and to forth a compromise between systematical cost and BEV owner cost, in conjunction with intelligent thermal management to avoid charging at lower temperatures.

This work provides guidance for priority selection between BEVs and FCEVs to meet the necessary flexibility requirements of the VRE system with the least system cost, in line with the current society, technology development, and policies. The effects of temperature and charging durations on BEV flexibility value are separate. How the coupling effects of the two factors are not involved in this work. A future work is to develop models and algorithms that integrate temperatures and charging times to evaluate EV flexibility values. In addition, investigation on the depth of battery charging and discharging and its influence on BEV flexibility is considered as another future work.

## Methods
### DOLPHYN model
The DOLPHYN model[62] (see "Code Availability") simultaneously optimizes infrastructure investments and operations across both electricity and $H_2$ supplies to determine the lowest total system cost to meet electricity and $H_2$ demands. Optimization is achieved while adhering to a variety of technological and system operation constraints enforced over representative periods (weeks in this study) at an hourly resolution as well as policy factors such as $CO_2$ prices. The DOLPHYN model can be configured to simulate the deployment of a variety of generation technologies, storage technologies, and transmission to meet the hourly electricity and $H_2$ demands in each defined zone over the modeled representative periods. The developed model can incorporate a wide range of power and $H_2$ technology options, including VRE generation, carbon capture and storage (CCS) applied to power and $H_2$ generation, and trucks (gaseous and liquid) and pipelines for $H_2$ transportation. Power systems and the $H_2$ supply chain are coupled through electrolysis and power generation technologies fueled by $H_2$, as well as electricity consumption in $H_2$ compression/liquefaction.

The operational constraints of the model, implemented at an hourly resolution, include a) the supply-demand balance for $H_2$ and electricity in each zone, b) inventory balance constraints for stationary storage technologies, c) inventory balance constraints related to trucks at a given location (any of the zones and routes, including arriving, departing or in transit trucks) and considering different states (empty and full), and d) the linearized unit commitment for conventional thermal power generation technologies and natural gas-based $H_2$ production technologies. We model these operational constraints at an hourly resolution over a set of representative weeks selected by applying time-series clustering to annual demand and VRE resource profile data to approximate annual system operations. Time-domain reduction preserves the chronological variability of energy demands and VRE resource availability, as well as the correlations among them, while reducing the model size to remain computationally tractable. Process-level $CO_2$ emissions are penalized with a price on emissions in both sectors.

A greenfield 2050 system with the exception of existing interzonal transmission, hydropower generation (both domestic and imported from Canada) and pumped hydro storage capacity in the US northeast was modeled in this study. More details on the system settings can be found in He et al.[63].

Considering the fact that some of the parameters are integer in nature, the optimization problem should be solved by mixed integer

programming (MIP). However, our model size is too large to be optimized with high efficiency using MIP. Hence, to improve computational tractability, we used prudent linearization to accelerate problem solving, which was validated in our previous work[49,63].

## Flexibility value calculation

The flexibility values of BEVs and FCEVs are calculated based on the objective values of the DOLPHYN model. The minimum system costs under the scenarios without flexible EV charging ($Y_0$), with flexible BEV charging considering the BEV degradation cost ($Y_{BEV}$), with flexible BEV charging but without considering the BEV degradation cost ($Y_{BEVND}$), and with flexible FCEV charging ($Y_{FCEV}$) are defined as follows:

$$Y_0 = \min_{x \in \Phi_0} \left[ C_{CAP}(x) + C_{OP}(x) + C_{EM}(x) \right] \tag{1}$$

$$Y_{BEVND} = \min_{x \in \Phi_{BEV}} \left[ C_{CAP}(x) + C_{OP}(x) + C_{EM}(x) \right] \tag{2}$$

$$Y_{BEV} = \min_{x \in \Phi_{BEV}} \left[ C_{CAP}(x) + C_{OP}(x) + C_{EM}(x) + C_{DEG}(x) \right] \tag{3}$$

$$Y_{FCEV} = \min_{x \in \Phi_{FCEV}} \left[ C_{CAP}(x) + C_{OP}(x) + C_{EM}(x) \right] \tag{4}$$

where $\Phi_0$, $\Phi_{BEV}$, and $\Phi_{FCEV}$ denote the operational and policy constraints of DOLPHYN including no flexible EV charging constraints, flexible BEV charging constraints, and flexible FCEV charging constraints, respectively, $C_{CAP}$, $C_{OP}$, $C_{EM}$, and $C_{DEG}$ are the functions of the capital cost, operational cost, emission cost, and BEV degradation cost, respectively, and $x$ denotes the decision variables of DOLPHYN, including planning and scheduling variables of the various resources in electricity and hydrogen supply chains.

The flexibility values of FCEVs and BEVs with and without considering BEV degradation, indicated by $V_{FCEV}$, $V_{BEV}$, and $V_{BEVND}$, are calculated as the differences between the minimum system cost without flexible charging ($Y_0$) and the corresponding minimum system cost with flexible charging:

$$V_{BEVND} = Y_0 - Y_{BEVND} \tag{5}$$

$$V_{BEV} = Y_0 - Y_{BEV} \tag{6}$$

$$V_{FCEV} = Y_0 - Y_{FCEV} \tag{7}$$

It is noted that the flexibility values of BEVs and FCEVs are analyzed preconditioned on the existence of BEV and FCEV fleet and corresponding charging/refueling infrastructure in the future. The cost of hydrogen refueling infrastructure depends on the total FCEV refueling demand, rather than the flexible part of the refueling demand. From a marginal perspective we used in this study, flexibly operating the refueling infrastructure does not directly add to the refueling infrastructure capacity or cost. Hence, the cost of hydrogen refueling infrastructure is not a significant factor (or cost) when assessing the flexibility of FCEVs, since this cost is fixed and insusceptible to the flexible scheduling of FCEVs.

## Degradation cost calculation

The degradation cost of the BEV battery, $C_{DEG}$ ($/kWh), is calculated by dividing the capital/replacement cost of the battery, $C_{bat}$ ($/kWh), by the cycle life $L_{bat}$ (Supplementary Table 6), as follows:

$$c_{DEG} = \frac{C_{bat}}{L_{bat}} \tag{8}$$

Under the various base electrode thickness and porosity scenarios, we assume a $200/kWh capital cost of battery replacement[25]. In the other cases, we calculate an additional unit capacity battery capital cost based on the required extra active materials, the price of active materials, and the battery capacity change (Supplementary Table 5).

Extra degradation of FCEV flexible charging was not considered since the hydrogen quality and refueling pressure of FCEVs are fixed and unaffected by these variables. The demand deferral shifts the FCEV refueling time only; it does not shorten or extend it. The lifetimes of electrolyzers and fuel cells are simplified as a fixed parameter (e.g., the lifetime of the fuel cell is approximated as 10 years), as shown in Supplementary Table 2.

## Flexible charging modeling

We model EV flexible charging as the deferrable demand, which can be delayed after arrival. In other words, the deferrable demand is the unserved demand in the following time window. It is readily comprehensible for BEV charging. Under this condition, we assume the EV chargers are enough and available. FCEV refueling deferral can be achieved through mechanism design, such as time-variant hydrogen prices or coupons[64]. As the prices of gas stations can be checked and compared on apps like Google Maps, it is easy to provide incentives for refueling deferral by sending customers the hydrogen refueling price information, including the potential cost savings of refueling deferral, through such apps. Although a vehicle might not be at the refueling station, the FCEV refueling can be scheduled through the interaction between the APPs and the vehicle with the development of the internet of vehicles. From a marginal perspective, there are always some vehicles in a large EV fleet that need to be charged or refueled. Consequently, the demands for FCEV refueling and BEV charging exist at all times, whether on the road or in parking lots. The potential incentives or programs for drivers as well as their willingness to shift BEV charging and FCEV refueling are similar despite the difference in BEV charging time and FCEV refueling time[64,65]. Drivers should tend to charge or refuel EVs during the period when the electricity or hydrogen prices are at their lowest within a time window (1 h or 6 h in this work). The time window represents the longest time the EV charging/refueling demands can be shifted, or in other words, the maximum time EVs can wait to be fully charged or refueled[66]. Therefore, the time windows for FCEV refueling and BEV charging are treated similarly.

The maximum deferrable demand is a fraction of the available capacity in a particular time step. For the marginal perspective, the maximum scale of deferrable EV demand for each zone is kept within a relatively small range (0 to 20 tonne $H_2$ per hour or equivalent MW in LHV, at a step of 5) in our analysis.

The total cumulated deferred charging demand at time $t$, denoted by $x_{y,z,t}^{FC\_S}$, can be obtained by deducting the served charging demand $x_{y,z,t}^{FC\_C}$ from and adding the delayed charging demand $x_{y,z,t}^{FC\_D}$ to the cumulated deferred charging demand in the last time step as:

$$x_{y,z,t}^{FC\_S} = x_{y,z,t-1}^{FC\_S} - x_{y,z,t}^{FC\_C} + x_{y,z,t}^{FC\_D} \tag{9}$$

where $y$ and $z$ denote the vehicle type and location of the EV charging demand, respectively. The flexible charging of either BEVs or FCEVs is characterized by the maximum deferrable demand capacity at a particular time step $t$, $D_{y,z,t}^{max}$, and the maximum time this demand can be delayed, defined by parameter $T_{y,z}$, as follows:

$$x_{y,z,t}^{FC\_D} \leq D_{y,z,t}^{max} \tag{10}$$

$$\sum_{\tau = t+1}^{t+T_{yz}} x_{y,z,\tau}^{FC\_C} \geq x_{y,z,t}^{FC\_S} \tag{11}$$

In this work, $T_{y,z}$ is set to one or six hours, and $D_{y,z,t}^{\max}$ ranges from 0 to 20 tonne H$_2$/hour or equivalent MW in LHV. The above formulations are part of $\Phi_{\text{BEV}}$ and $\Phi_{\text{FCEV}}$, not $\Phi_0$.

The electricity demand data are based on 2018 NREL electrification futures study load projection for 2050[67]. While the H$_2$ demands are developed based on available fuel consumption data, hourly refueling profiles, and the relative penetration of FCEVs[68] (Supplementary Fig. 3).

## Hydrogen supply chain and balance constraints

The model for hydrogen supply chain scheduling is involved as an essential part of the whole DOLPHYN model, incorporating all steps in the hydrogen supply chain including hydrogen production, compression, transmission, and storage, as shown in Supplementary Fig. 2. Almost all critical technological options are considered in each step.

The total cost of hydrogen supply chain involves the cost of hydrogen generation, conversion, transmission, and storage. For hydrogen production, electrolyzer, SMR with and without CCS are modeled. Regarding hydrogen transmission, gas/liquid trucks and pipelines for flexible transmission are modeled.

The hydrogen supply chain is scheduled following the hydrogen balance constraint. For a specific zone at a moment, the amount of H$_2$ production $h_{z,t}^{\text{GEN}}$ plus the amount of transported H$_2$ (positive for imports) $h_{z,t}^{\text{TRA}}$ and the amount of H$_2$ discharged from storage $h_{z,t}^{\text{DIS}}$ should be equal to the amount of H$_2$ charged to storage $h_{z,t}^{\text{CHA}}$ plus the H$_2$ demand $D_{z,t}$ and minus the lost demand $h_{z,t}^{\text{LOS}}$.

$$h_{z,t}^{\text{GEN}} + h_{z,t}^{\text{TRA}} + h_{z,t}^{\text{DIS}} = h_{z,t}^{\text{CHA}} + D_{z,t} - h_{z,t}^{\text{LOS}} \tag{12}$$

## Porous electrode theory-based battery degradation model

The PET model is the most widely used first-principles electrochemical model that describes many of the physicochemical details of lithium-ion battery dynamics[58,69]. In the PET model, each porous electrode contains an electrically conductive solid phase in close contact with a liquid electrolyte, and the two phases are coupled via interfacial electrochemical kinetics[58,70]. Lithium ions are dynamically transported between the active particles in the electrolyte. Specifically, the main governing equations of the PET model for charge conservation in solid electrodes and the electrolyte can be expressed as follows:[69,70]

$$\nabla \cdot (\sigma_s^{\text{eff}} \nabla \phi_s) = j \tag{13}$$

$$\nabla \cdot \left( k_e^{\text{eff}} \nabla \phi_e \right) + \nabla \cdot \left( k_e^{\text{eff}} \nabla \ln c_e \right) = -j \tag{14}$$

where $\sigma_s^{\text{eff}}$ is the effective conductivity of the electrodes, $k_e^{\text{eff}}$ is the effective kinetic rate, $c_e$ is the concentration of the electrolyte, $\phi_s$ and $\phi_e$ are the solid and electrolyte potentials, respectively, and $j$ is the volumetric current density.

The governing equations for species conservation of the electrolyte and active material particles can be expressed as:[69,70]

$$\frac{\partial(\varepsilon c_e)}{\partial t} = \nabla \cdot \left( D_e^{\text{eff}} \nabla c_e \right) + \frac{1 - t_+}{F} j \tag{15}$$

$$\frac{\partial c_s}{\partial t} = \frac{1}{r^2} \frac{\partial}{\partial r} (D_s^{\text{eff}} r^2 \frac{\partial c_s}{\partial r}) \tag{16}$$

where $c_s$ is the concentration of solid particles, $t_+$ is the transference number, $F$ is Faraday's constant, $r$ is a one-dimensional spatial variable, and $D_e^{\text{eff}}$ and $D_s^{\text{eff}}$ are the effective diffusion coefficients of the electrolyte and electrodes, respectively.

In terms of battery degradation modeling, this work uses the same degradation modeling approach as that of Yang et al.[70], which considers lithium plating as side reactions in the anode. Therefore, two electrochemical reactions occur in the anode, and the volumetric current density $j$ comprises the transfer current density of lithium intercalation and lithium deposition reactions.

The key parameters of the above electrochemical model used for lithium-ion batteries with chemistries of lithium-manganese-cobalt-oxide (NMC)/graphite in this work can be found in the Base column of Supplementary Table 5. This work uses PETLION[58]—a Julia implementation of the above electrochemical model based on the finite volume method—as a battery simulator to compute the degradation cost of BEVs considering different charging protocols. The cells are simulated to be charged from 30% state of charge (SOC) to 80% SOC using different charging protocols under various time constraints. More details on the PET-based electrochemical model and its software implementation can be found in Berliner et al.[58] and Fuller et al.[69].

Notably, the PET model here is employed for battery cells. To obtain the lifetime of battery modules, a conversion coefficient[56] is adopted. This coefficient can be calculated based on multiphysical modeling involving electrochemical and series-parallel circuit models to obtain the relationship between the cycle life of battery cells and modules. The conversion coefficient used in this work is 0.44. More details can be found in Xia et al.[71].

## Data availability

The authors declare that the data supporting the findings of this study are available within the paper and its supplementary information files.

## Code availability

The code for the macro-level DOLPHYN model generated in this study is available at https://github.com/macroenergy/DOLPHYN /. The code for the micro-level PET-based model to simulate battery degradation and obtain battery lifespan is available at https://cloud.tsinghua.edu.cn/d/f97ff4edb7384be49e13.

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

## Acknowledgements

G.H. was supported by National Key Research and Development Program of China under grant number 2022YFB2405600. B.J. and G.H. were supported by the National Natural Science Foundation of China under grant numbers 62273197, 62192751, 72271008, and 72342004. B.J. was supported by the Beijing Natural Science Foundation under grant number L233027, the National Key Research and Development Program of China under grant number 2022YFE0197600, and the Beijing National Research Center for Information Science and Technology under grant number BNR2023RC01008.

## Author contributions

G.H. and B.J. conceived the research. R.L., G.H., and X.W. performed the modeling and numerical experiments. R.L., G.H., X.W., D.M., H.Z., Y.S.H., and B.J. interpreted the results. All authors edited and reviewed the manuscript. G.H., Y.S.H., and B.J. supervised the work.

## Competing interests

The authors declare no competing interests.
