## [Peer Review File · Nature Communications]

A Cross-Scale Framework for Evaluating Flexibility Values of Battery and Fuel Cell Electric VehiclesReviewers' comments:

Reviewer #1 (Remarks to the Author):

Recommendation:

I highly recommend this article for acceptance, with minor revisions

Overall comments:

I find the article reads well and is extremely interesting. To be honest, I am rather envious, as this is the work I dreamed of doing. I commend the authors on their thorough and insightful approach and analysis. This is very timely work, with the extremely important finding that FCEVs can be better flexibility providers under certain scenarios. The quote "In the future, if the battery lifetime performance is improved by twofold and the battery replacement cost is reduced by half, ..." sets very important and useful targets for the battery industry.

The paper provides an important method to inform the transport electrification strategies for different countries, based on their ambient temperatures, as well as highlighting the essential role of policy in reducing carbon emissions.

The methodology is very clearly explained. Their approach stands out from other studies by tackling the coupling between electricity and transportation and is unique in using a high temporal resolution, multiscale model, including electricity and hydrogen supply chain technologies, as well as including the BEV degradation cost of mixing charging strategies.

There is no micro-level model for the FCEVs, while this is not necessary, it builds in an assumption that there is no degradation in FCEVs/electrolysers of any kind. It is an acceptable compromise, because the focus is on fast charging, but for future work, other mechanisms could be included? The authors explain this issue fully on page 9, so no alteration is required.

I would be extremely interested to see how this work expands, using a broader range of use cases, perhaps mixing fast charging with longer duration charging, to determine the maximum number of times fast charging can be allowed, for example? The focus here is on fast-charging at low temperatures; would be interesting to see other mechanisms, for example high temperature degradation.

My limitations as a reviewer

I have not done whole energy systems flexibility modelling myself; my focus is on sustainability and battery lifetime extension.

I give detailed comments and suggestions for improvement in the attached word document.

Reviewer #2 (Remarks to the Author):

The work evaluates the economic value of flexible charging of electric vehicles or refueling of fuel cell

vehicles for the energy system as a whole. This is a way to quantify the impact of demand side management with two new and potentially significant energy demands. This is an important topic area. The work contributes to existing knowledge by using a coupled energy system optimization model along with a model of battery degradation in a novel way. In particular, the integrating estimates of battery degradation specific to the modeled operations, temperature and speed of charging, provides a rationale estimate to the flexibility value that is not possible without this method.

There are issues with the work that are large enough to discourage publication in Nature Communications. First, focusing on comparing electric vehicles and fuel cell vehicles detracts from the value of the work. The conception of fuel cell vehicle refueling is erroneous. The authors have modeled FCV refueling as a dispatchable demand based on when a vehicle is parked. FCVs will be refueled at stations with specialized equipment and in short durations like gasoline vehicles. This does not lend itself to deferring demand until later. This makes the FCV portion of the work disconnected from reality and of little value. The flexibility value of the hydrogen sector would be a more compelling comparison. Other concerns are listed below:

- The DOLPHYN model is insufficiently described in the text and supplemental information. Figure 1 is great but the text does not describe the underlying method at all leading the reader to chase down the code to find that it is a cost minimizing MIP with an annual cost basis and with a limited number of regions to capture space.

- Capturing the cost of a hydrogen refueling infrastructure accurately requires higher spatial resolution or the use of heuristics that are not documented.

- The timing of charging demand is described as within a given time from parking the EV. What is used to create the distribution of vehicle parking times and whether chargers are available for a given parking session?

- The logic of selection of cases to be evaluated is also not clear. The time constraints of the charging events of the different cases were muddled in the text, much clearer as a table in the Appendix.

- The results switch metrics from percent cost savings to absolute cost savings starting at section 2.2. This makes the relative impact of the sensitivities evaluated in 2.2 and 2.3 impossible to compare with the results in 2.1.

Reviewer #3 (Remarks to the Author):

The authors' report on a flexibility value for Battery and Fuel Cell Electric Vehicles is certainly an interesting proposal. However, the reviewer does not feel strongly that this paper offers the sort of conceptual advance or striking demonstration of practical capabilities.

The main conclusions of the work are relatively intuitive and do not present novel or counter-intuitive arguments. The main conclusion is a common consensus in the research field, such as that battery aging considerations at the operational level limit the depth of discharge and therefore limit flexibility.

Studies similar to the model proposed by the authors are already abundant, such as the comparison of FCEVs and BEVs incorporating Variability in a Life Cycle Assessment framework. I'm not sure if there is a term for the 'flexibility value' that the authors emphasize, and it is a type of cost as well. There is no obvious innovation in such a definition compared to Life Cycle Cost.

Both the supplementary material and the methods section lack a discussion of the core model, a derivation of the computational process, or an elaboration of the algorithm design. The methodology and its innovation need more justification. It is unclear how the authors applied and customized the Decision Optimization for Low Carbon Power and Hydrogen Nexus model, and reproducing the results is therefore difficult. The unit cost calculation for battery degradation is also unclear, how did the result of Supplementary Table 6 come?

The interpretation of the figures is suggested to be clearer and more detailed, e.g., Fig 1a. The four scenarios are unknown in this figure, why the charging costs are four parallel straight lines, and what is the range of time dimensions considered?

How realistic are the assumptions stated in Section 2 and how do they lead to discrepancies when compared to "real world" events?

In Section 3, the authors designed a large number of scenarios, but the introduction of the scenarios and the definition of key attributes are vague. For example, fast and normal charging, deferrable demand, charging duration, etc.

Response to Reviewer #1:

Review Comment 0: This is very timely work, with the extremely important finding that FCEVs can be better flexibility providers under certain scenarios. The quote “In the future, if the battery lifetime performance is improved by twofold and the battery replacement cost is reduced by half, ...” sets very important and useful targets for the battery industry.

The paper provides an important method to inform the transport electrification strategies for different countries, based on their ambient temperatures, as well as highlighting the essential role of policy in reducing carbon emissions.

The methodology is very clearly explained. Their approach stands out from other studies by tackling the coupling between electricity and transportation and is unique in using a high temporal resolution, multiscale model, including electricity and hydrogen supply chain technologies, as well as including the BEV degradation cost of mixing charging strategies.

Author Response: We thank reviewer 1 for recognizing the novelty and quality of this manuscript.

Review Comment 1:

There is no micro-level model for the FCEVs, while this is not necessary, it builds in an assumption that there is no degradation in FCEVs/electrolysers of any kind. It is an acceptable compromise, because the focus is on fast charging, but for future work, other mechanisms could be included? The authors explain this issue fully on page 9, so no alteration is required.

Author Response: We assumed an operational constraint that the power inputs of electrolyzers are limited under the rated power capacities. Then producing hydrogen does not incur notable extra degradation. In addition, fueling time of FCEVs is usually short enough for the time windows considered in this study. Thus, flexible fueling of FCEVs in both normal-charging and fast-charging setting will not cause extra degradation cost. Therefore, the degradation in FCEVs/electrolysers is not considered in this work. For the future work, other mechanisms are expected to be included. For instance, the degradation or maintain cost of FCEVs/electrolysers caused by the membrane chemical degradation or mechanical degradation.

Review Comment 2:

I would be extremely interested to see how this work expands, using a broader range of use cases, perhaps mixing fast charging with longer duration charging, to determine the maximum number of times fast charging can be allowed, for example? The focus here is on fast-charging at low temperatures; would be interesting to see other mechanisms, for example high temperature degradation.

Author Response:

- Thank you for your valuable comment. It is a good idea to continue the study on determining the maximum number of fast-charging times in terms of systematical cost, user cost, or flexibility value. We will consider it seriously.
- Existing studies on battery degradation show that fast-charging at low temperatures will definitely accelerate battery degradation. So, we focus on low temperatures here. Other mechanisms such as high temperature, discharging or operation conditions could be also considered in the future work.

Review Comment 3:

I give detailed comments and suggestions for improvement in the attached word document.

Author Response:

- Thank you for your detailed comments. The text “*FCEVs outperform BEVs in more cases*” in Section 2.3 is under the precondition “*When the temperature is reduced to 0 °C*”.
- In the first sentence of Section 5.4, “*after arrival*” means after BEVs arrive chargers.
- We have added/revised the manuscript according to your comments, by
 - (1) Revising “*by considering*” to “*after considering*” in the Abstract.
 - (2) Adding “*optimization*” in the Abstract.
 - (3) Adding the text in the Abstract “*With advanced technologies applied in the future, a twofold extended battery lifetime is expected to improve BEV flexibility by additionally reducing about 4% system cost.*”.
 - (4) Adding the text at the end of Abstract “*The proposed cross-scale methodology has broad implications for the assessment of emerging energy technologies with complex dynamics and the corresponding low-carbon transition pathway.*”.
 - (5) Revising “*to the energy system*” to “*for the energy system's*” in the second sentence of the first paragraph of Introduction section.
 - (6) Adding “*demand-side*” before “*response*” in the fourth sentence of the first paragraph of Introduction section.
 - (7) Revising “*produce many potential flexibility providers*” to “*produce a distributed source of flexibility*” in the fifth sentence of the first paragraph of Introduction section.
 - (8) Deleting “*notably*” in the last sentence of the first paragraph of Introduction section.
 - (9) Revising “*transportation electrification*” to “*the electrification of transport*” in the first sentence of the second paragraph of Introduction section.
 - (10) Revising “*show advantages at larger driving ranges*” to “*have longer driving ranges*” in the second sentence of the second paragraph of Introduction section.
 - (11) Adding “*limited*” before “*longevity*” in the third sentence of the second paragraph of Introduction section.
 - (12) Revising “*the scale effect*” to “*the increasing scale of*” in the fifth sentence of the second paragraph of Introduction section.
 - (13) Deleting “*alternative*” in the sixth sentence of the second paragraph of Introduction section.
 - (14) Adding “*(involving any taxes imposed)*” after “*emission cost*” in the second sentence of the fourth paragraph of Introduction section.
 - (15) Revising “*provide greater flexibility value*” to “*providing greater flexibility value in the battery's subsequent service life*” in the last sentence of the fourth paragraph of Section 2.
 - (16) Adding the text to link between H₂ demand and decarbonization level at the last paragraph of Section 2 “*The parameters of the scenarios reflect the decarbonization progress. For instance, the deeper decarbonization process might show higher hydrogen demand, larger EV penetration (denoted as deferrable demand in this work), and larger carbon prices.*”.
 - (17) Adding the definition of “*deferrable demand*” and “*LHV*” at the end of caption of Figure 2: “*Deferrable demand here signifies the flexible consumption of hydrogen or electric power for EVs. Larger deferrable demand means a larger level of EV penetration. LHV is short for low heating value.*”.

- (18) Revising “pink” to “yellow” in caption of Figure 3.
- (19) Deleting the last sentence of caption of Figure 3: “*The colors from red to blue indicate gradually improved flexibility values*”.
- (20) Adding “variation of” before “system cost” in caption of Figure 4.
- (21) Revising “the *electrolytic-only pathway and the mixed hydrogen pathway*” to “the *mixed hydrogen pathway and the electrolytic-only pathway*” in caption of Figure 4.
- (22) Deleting the text in the caption of Figure 4: “*with increasing deferrable EV demand.*”.
- (23) Revising “the *flexibility of BEVs than that of FCEVs*” to “the *flexibility from BEVs than that from FCEVs*” in the first sentence of the second paragraph of Section 2.2.
- (24) Revising “*pathway is remarkably*” to “*pathway are remarkably*” in the second sentence of the second paragraph of Section 2.2.
- (25) Revising “*The high sensitivity of the BEV flexibility value to the hydrogen pathway occurs because of the stronger coupling of the power system and hydrogen supply chain and the substitution effect of the flexible hydrogen supply chain.*” to “*The high sensitivity of the BEV flexibility value to the hydrogen pathway can be explained by the stronger coupling of the hydrogen supply chain to the power system, through electrolysis, and the substitution effect of the flexible hydrogen supply chain to BEVs.*” In the third sentence of the second paragraph of Section 2.2.
- (26) Revising “*green-only*” to “*electrolytic-only*” in the second sentence of the first paragraph of Section 2.3.
- (27) Revising “*increased corresponding*” to “*corresponding increased*” in the last sentence of the first paragraph of Section 2.3.
- (28) Revising “*The higher H₂ demand, the greater potentials for FCEVs in terms of flexibility*” to “*The higher H₂ demand increases the potential for FCEVs to provide flexibility*” in the fourth sentence of the second paragraph of Section 2.3.
- (29) Adding a reference to prove the second sentence in the second paragraph of Section 3
- (30) Deleting “*light-colored*” and “*deep-colored*” in the caption of Figure 6.
- (31) Add a review reference to the last sentence of the caption of Figure 6.
- (32) Revising “*green hydrogen-only*” to “*electrolytic hydrogen-only*” in the fifth sentence of the second paragraph of Conclusions.
- (33) Adding the text in the section of Conclusions “*on account of the stronger coupling of the hydrogen supply chain to the power system through electrolysis*”.
- (34) Adding the text at the end of the third paragraph “*to \$50 kWh*”.
- (35) Revising “*batteries of BEVs*” to “*BEV batteries*” in the Conclusions section.
- (36) Deleting the second sentence in the fourth paragraph of Conclusions section: “*Faster charging and a lower temperature can notably decrease the lifetime and BEV flexibility value because of the accelerated battery degradation.*”.
- (37) Adding the text at the end of the fourth paragraph of Conclusions section: “*and to forth a compromise between systematical cost and BEV owner cost,*”.

Response to Reviewer #2:

Review Comment 0: The work evaluates the economic value of flexible charging of electric vehicles or refueling of fuel cell vehicles for the energy system as a whole. This is a way to quantify the impact of demand side management with two new and potentially significant energy demands. This is an important topic area. The work contributes to existing knowledge by using a coupled energy system optimization model along with a model of battery degradation in a novel way. In particular, the integrating estimates of battery degradation specific to the modeled operations, temperature and speed of charging, provides a rationale estimate to the flexibility value that is not possible without this method.

Author Response: Thank you for your comments and approval of our method.

Review Comment 1:

There are issues with the work that are large enough to discourage publication in Nature Communications. First, focusing on comparing electric vehicles and fuel cell vehicles detracts from the value of the work. The conception of fuel cell vehicle refueling is erroneous. The authors have modeled FCV refueling as a dispatchable demand based on when a vehicle is parked. FCVs will be refueled at stations with specialized equipment and in short durations like gasoline vehicles. This does not lend itself to deferring demand until later. This makes the FCV portion of the work disconnected from reality and of little value. The flexibility value of the hydrogen sector would be a more compelling comparison.

Author Response:

- While we agree that the mechanisms to utilize the flexibility of BEVs and FCEVs are different, modelling the refueling of FCEVs as a dispatchable demand is reasonable from the two perspectives. (1) First, the flexibility value of FCEVs we analyze in this study is indeed attached to the flexibility of the hydrogen supply chain, specifically the storage capacities at the hydrogen refueling stations. Such flexibility is preconditioned on the deployment of FCEV fleet, and therefore, we attribute this flexibility to the FCEV pathway. (2) Second, refueling deferral can be achieved through mechanism design, such as time-variant hydrogen prices or coupons. As the prices of gas stations can be checked on apps like Google Maps and compared, it is easy to provide incentives for refueling deferral by sending customers the hydrogen refueling price information, including the potential cost savings of refueling deferral, through such apps. While deferring gasoline refueling is also possible, the value is not as high as that for hydrogen, which will be largely produced by intermittent renewable resources.
- We have added/revise the manuscript by
 - (1) Inserting the below text into the sixth paragraph of Section 2 to state the mechanisms of FCEVs flexibility analysis: *“The mechanisms to utilize the flexibility of FCEVs are different from those of BEVs. Considering the fact that the flexibility value of FCEVs in this study is indeed attached to the flexibility of the hydrogen supply chain, specifically the storage capacities at the hydrogen refueling stations, we modeled the refueling of FCEVs as a dispatchable demand. Such flexibility is preconditioned on the deployment of the FCEV fleet, and therefore, we attribute this flexibility to the FCEV pathway.”*
 - (2) Adding the below text after the first sentence of Section 5.4 to further clarify the rationality of modeling FCEVs as deferrable demand: *“FCEVs refueling deferral can be achieved*

through mechanism design, such as time-variant hydrogen prices or coupons. As the prices of gas stations can be checked on apps like Google Maps and compared, it is easy to provide incentives for refueling deferral by sending customers the hydrogen refueling price information, including the potential cost savings of refueling deferral, through such apps.”.

Review Comment 2:

The DOLPHYN model is insufficiently described in the text and supplemental information.

Figure 1 is great but the text does not describe the underlying method at all leading the reader to chase down the code to find that it is a cost minimizing MIP with an annual cost basis and with a limited number of regions to capture space.

Author Response:

- Thank you for your comment. We described the DOLPHYN model in “Methods” section with the operational constraints included. We have revised the manuscript to enrich the description on the model by
 - (1) Inserting the text into the first paragraph of Section 2 to describe the DOLPHYN model *“The DOLPHYN is a sector-coupling planning and decision optimization model to minimize the cost of the low-carbon power network (see Methods section). It optimized the costs to identify the most effective and efficient design and operation of the energy system, by modeling the coupling and conversion of different energy vectors and revealing the competition and complementary among different technologies.”.*
 - (2) Adding a sentence at the end of the first paragraph of Section 2: *“Linear programming solved by Gurobi with barrier methods⁶² was applied to address the optimization problem under four operational constraints (detailed constraints are available in Methods).”.*
 - (3) Adding a paragraph after the first paragraph of Section 2 to further describe the method according to Figure 1: *“As shown in Fig. 1a, we customized the DOLPHYN model to optimize the energy systems without and with flexible charging, taking the optimized cost of the former one as the benchmark cost. The flexibility value of EVs was obtained by the difference value between the benchmark cost and the system least-cost with EV flexible charging. With regard to BEVs, the DOLPHYN model was also customized to the energy system involving battery degradation cost. Then the flexibility value of BEVs considering battery degradation was computed by subtracting the system least cost with flexible charging and degradation from the benchmark cost. More details are available in Methods and Supplementary Figure 1.”.*
 - (4) Adding a paragraph at the end of Section 5.1 to state the solving method *“Considering the fact that some of the parameters are integer in nature, the optimization problem should be solved by mixed integer programming (MIP). However, our model size is too large to be optimized with high efficiency using MIP. Hence, to improve computational tractability, we used prudent linearization to accelerate problem solving, which was validated to improve computational efficiency in our previous work^{62,75}.”.*

Review Comment 3:

Capturing the cost of a hydrogen refueling infrastructure accurately requires higher spatial resolution or the use of heuristics that are not documented.

Author Response:

- The cost of hydrogen refueling infrastructure depends on the total FCEV refueling demand, rather than the flexible part of the refueling demand. From a marginal perspective we used in this study, flexibly operating the refueling infrastructure does not directly add to the refueling infrastructure capacity or cost. In this work, we analyzed the flexibility of BEVs and FCEVs preconditioned on the existence of BEV and FCEV fleet and charging/refueling infrastructure in the future. Since the cost of hydrogen refueling infrastructure is fixed and unsusceptible to the flexible scheduling of FCEVs, it is not a significant factor (or cost) when assessing the flexibility values of FCEVs.
- We have revised the manuscript to enhance the clarification on the model by
 - (1) Inserting the below text in the last paragraph of Section 5.2 to state the reason why we did not consider the cost of hydrogen refueling infrastructures: *“It is noted that the flexibility values of BEVs and FCEVs are analyzed preconditioned on the existence of BEV and FCEV fleet and corresponding charging/refueling infrastructure in the future. The cost of hydrogen refueling infrastructure depends on the total FCEV refueling demand, rather than the flexible part of the refueling demand. From a marginal perspective we used in this study, flexibly operating the refueling infrastructure does not directly add to the refueling infrastructure capacity or cost. Hence, the cost of hydrogen refueling infrastructure is not a significant factor (or cost) when assessing the flexibility of FCEVs, since this cost is fixed and unsusceptible to the flexible scheduling of FCEVs.”*

Review Comment 4:

The timing of charging demand is described as within a given time from parking the EV. What is used to create the distribution of vehicle parking times and whether chargers are available for a given parking session?

Author Response:

- Our analysis is based on a marginal perspective, which means that the flexible charging demand is less than the total EV charging demand at the times when flexible charging demand is needed. This is why the scale of deferrable EV demand is kept within a relatively small range in our analysis. We take this assumption because the future EV charging distribution and the ratio of deferrable demand are highly uncertain and may deviate from the current patterns. Only for the “BEV Average” scenario, we assume BEV arrivals are uniformly distributed within an hour, so that we can average the extra degradation cost of 15-min, 30-min, 45-min and 60-min charging duration.
- Under the marginal perspective, the chargers are also not a bottleneck. The chargers are assumed to be enough and available for BEV charging, since available chargers can be checked through some APPs, Google Map for instance. Besides, the EV flexible charging is modeled as the deferrable demand. It means that BEVs do not need to be charged immediately after their arrivals, as stated at the first sentence of Section 5.4.
- We have added/revised the manuscript by

- (1) Adding a paragraph as the third paragraph of Section 2 to explain the deferrable demand and marginal perspective: *“We modeled the EV flexible charging as deferrable demand, which is the flexible consumption of hydrogen or electric power for EVs. It implies that the EVs do not need to be charged or refueled immediately after their arrivals. Our study on flexibility assessment of EVs was based on a marginal perspective, which means that the flexible charging demand is less than the total EV charging demand at the times when flexible charging demand is needed. Thus, the scale of deferrable EV demand was kept within a relatively small range in our analysis. We took this assumption because the future EV charging distribution and the ratio of deferrable demand are highly uncertain and may deviate from the current patterns.”*.
- (2) Adding the text after the second sentence of the third paragraph of Section 2 to clear the assumptions: *“It is assumed that the chargers are enough and available for BEV charging preconditioned on modeling EV flexible charging as deferrable demand.”*.

Review Comment 5:

The logic of selection of cases to be evaluated is also not clear. The time constraints of the charging events of the different cases were muddled in the text, much clearer as a table in the Appendix.

Author Response:

- Thank you for your valuable comment. We total selected five scenarios to compare flexibility values of BEVs and FCEVs under various charging modes, hydrogen pathways, and service temperatures. Two charging modes were evaluated, i.e., fast charging and normal charging. For fast charging, the charging load may be shifted to any time within one-hour window starting from vehicle arrival. Two cases namely “BEV Average” and “BEV Extreme” are involved in the fast-charging setting. In the former case, we use the average of the extra degradation costs associated with charging times of 15, 30, 45, and 60 min over the 60-min benchmark. In the latter case, we assume all BEVs are charged within 10min and use the extra degradation cost over the benchmark level.
- To improve the clarification, we have added/revised the manuscript by
 - (1) Adding the text at the end of the sixth paragraph of Section 2 to describe the case selection logic: *“Some parameters of the scenarios reflect the decarbonization progress. For instance, the deeper decarbonization process might show higher hydrogen demand, larger EV penetration (signified as deferrable demand in this work), and larger carbon prices. To model flexible EV charging, we considered two load-shifting settings at room temperature with the electrolytic-only hydrogen pathway applied. The charging duration of the fast-charging setting is less than 1 h, while that of the normal charging setting is 6 h). Different battery replacement costs from \$50/kWh to \$200/kWh are considered in the normal charging case. For the fast-charging case, an average charging scenario and an extreme fast-charging scenario are discussed. When comparing the flexibility values of BEVs and FCEVs under different hydrogen pathways or temperatures, the BEV average fast-charging setting with a \$100/kWh battery replacement cost is applied. Two hydrogen pathways include the electrolytic-only hydrogen and the mixed hydrogen pathway. Temperatures considered in this work are 25 °C, 10 °C, and 0 °C, respectively. Lower*

temperatures are discussed because studies have illustrated that lower temperatures evidently affect battery charging.”.

(2) The Supplementary Table 1 of the original version which summarizes all the scenarios for comparing flexibility values of BEVs and FCEVs has been moved into the manuscript. .

Table 1. Summaries of scenarios for comparing flexibility values of BEVs and FCEVs

Scenarios			Charging duration	Temperature	Battery replacement cost	Hydrogen pathway
Various charging modes	Fast charging	FCEVs (%)	1h	25°C	--	Electrolytic -only
		BEV Average (%)	15min,30min,45min,60min	25°C	\$100/kWh	
		BEV Extreme (%)	10min			
	Normal charging	FCEVs (%)	6h	25°C	--	
		BEV \$50-\$100 (%)			\$50/kWh - \$100/kWh	
		BEV \$100-\$150 (%)			\$100/kWh - \$150/kWh	
		BEV \$150-\$200 (%)			\$150/kWh - \$200/kWh	
Hydrogen pathway	Mixed	FCEVs (%)	1h	25°C	--	Mixed
		BEV Average (%)		25°C	\$100/kWh	
	Electrolytic-only	FCEVs (%)		25°C	--	Electrolytic -only
		BEV Average (%)		25°C	\$100/kWh	
Service temperature	FCEVs (%)		1h	25°C	--	Electrolytic -only
	BEV Average 25°C (%)			25°C		
	BEV Average 10°C (%)			10°C	\$100/kWh	
	BEV Average 0°C (%)			0°C		

Review Comment 6:

The results switch metrics from percent cost savings to absolute cost savings starting at section 2.2. This makes the relative impact of the sensitivities evaluated in 2.2 and 2.3 impossible to compare with the results in 2.1.

Author Response:

- Actually, the metrics used in Section 2.3 are the same as those in Section 2.1. Both are the percent cost saving. In Section 2.2, we compared some related factors (generation and storage capacities) to help explain the reasons when the electrolytic-only hydrogen pathway is adopted instead of the mixed hydrogen pathway, the BEV flexibility values obviously decreased, while FCEVs flexibility values are less affected. So, all these metrics in Section 2.2 shows as the absolute cost saving. We also offer corresponding results with percent cost saving in Supplemental Figure 1.
- As requested, we also modified Figure 4 in Section 2.2, with the system cost column changed into the form of percent cost saving.

Response to Reviewer #3:

Review Comment 0: The authors' report on a flexibility value for Battery and Fuel Cell Electric Vehicles is certainly an interesting proposal.

Author Response: Thank you for your comments. We respond to the concern on “Flexibility Value” versus “Life Cycle” first (comment 2) and then address the rest remarks in detail (comments 1 and 3-6).

Review Comment 2:

Studies similar to the model proposed by the authors are already abundant, such as the comparison of FCEVs and BEVs incorporating Variability in a Life Cycle Assessment framework. I'm not sure if there is a term for the 'flexibility value' that the authors emphasize, and it is a type of cost as well. There is no obvious innovation in such a definition compared to Life Cycle Cost.

Author Response:

- We have to respectively disagree with this review comment. Our work assesses the flexibility potentials that BEVs and FCEVs could provide for the whole variable renewable energy (VRE) systems in the future using a cross-scale approach including energy system optimization (ESO). While LCA has some overlaps with ESO in terms of the cost components, they are fundamentally different methods. LCA *calculates* the costs of a product or system (Oda et al, Renew. Sust. Energ. Rev., 159, 112214, 2022), while ESO *optimizes* the costs to identify the most effective and efficient design and operation of an energy system. ESO can model the coupling and conversion of different energy vectors and reveals the competition and complementary among different technologies. Therefore, ESO is more appropriate than LCA in comparing energy technology pathways especially when they are coupled and competing for resources, e.g., the coupling of power and hydrogen supply chains and the competing for renewable energy in the comparison of BEV and FCEV.
- Moreover, the 'flexibility value' is an entirely different definition from 'Life Cycle Cost'. The latter is the total cost generated during the life cycle of vehicles including production, usage, and liquidation (Li et al, Int. J. Hydrog. Energy, 46, 9553-9566, 2021), while 'flexibility' is defined as the ability of power systems to balance energy supply with demand in the low-carbon energy system (Cruz et al, Renew. Sust. Energ. Rev., 97, 338-353, 2018). It is a crucial concept for the VRE system by the reason of its uncertainty and intermittency. Considering the fact that less flexibility in the VRE system will increase the system cost, we quantify the EVs' flexibility value by computing the reduction of energy-system cost owing to EVs' flexible charging. In addition, to incorporate the battery degradation in the system cost, a micro-level model for degradation analysis is also integrated into the macro-level sector-coupling model, forming a cross-scale framework. The cross-scale methodology is pioneering in the incorporation of a sector-coupling macro-level model for energy system cost estimation and micro-level analyses of battery degradation.

As Reviewer 2 pointed out, “the integrating estimates of battery degradation specific to the modeled operations, temperature and speed of charging, provides a rationale estimate to the flexibility value that is not possible without this method.” The proposed cross-scale methodology also has broad implications for the optimization of emerging energy technologies

with complex dynamics and the corresponding low-carbon transition pathway. Therefore, our work and model are absolutely different from existing models for BEVs and FCEVs comparison.

- We have inserted the below text into the second paragraph of Introduction section of revised manuscript: “*Even though many existing studies have compared the BEVs and FCEVs in a life cycle assessment framework incorporating variability ⁴²⁻⁴⁵, the values of flexible charging of BEVs and FCEVs to the flexibility of the whole energy systems are still undiscovered and quantified.*”.

Review Comment 1:

However, the reviewer does not feel strongly that this paper offers the sort of conceptual advance or striking demonstration of practical capabilities. The main conclusions of the work are relatively intuitive and do not present novel or counter-intuitive arguments. The main conclusion is a common consensus in the research field, such as that battery aging considerations at the operational level limit the depth of discharge and therefore limit flexibility.

Author Response:

- Our work is pioneering in the quantitative estimation and comparison of the flexibility values of BEVs and FCEVs to the low-carbon energy system. Based on the quantitative study, our conclusions are novel in showing how much the flexibility value of BEVs has been inflated by not considering battery degradation. Almost existing studies on energy systems ignore battery degradation. According to our study, after involving the battery degradation cost in the total energy system cost, the flexibility reduction of BEVs is quantified by at least 4.7% of the minimum system cost and becomes comparable to that of FCEVs. The significance of future battery management technology improvements for mitigating battery degradation to offer greater BEV flexible potential is also quantified as a 4% additional VRE system cost reduction. The effects of BEV charging mode, policies including hydrogen pathway and carbon prices, and service temperatures on the comparative preference of BEVs and FCEVs in terms of their flexibility value are also quantitatively analyzed, providing guidance for priority selection between BEVs and FCEVs to meet the necessary flexibility requirements of the VRE system with the least system cost, in line with the current society, technology development, and policies. The above quantitative conclusions and implications to the transportation electrification industry would not be available without our cross-scale methodological framework integrating macro-level and micro-level models considering the battery degradation cost. In addition, the proposed cross-scale methodology has broad implications for the assessment of emerging energy technologies with complex dynamics and the corresponding low-carbon transition pathway.
- We have added/revised the manuscript by
 - (1) Revising the text in Abstract “*Fast charging and a low temperature environment could reduce the flexibility values of BEVs due to increased degradation.*” to be “*The flexibility reduction after considering battery degradation is quantified by at least 4.7% of the minimum system cost and enlarged under fast charging and low-temperature scenarios.*”.
 - (2) Adding the text in the last paragraph of Conclusion: “*In summary, this work provides guidance for priority selection between BEVs and FCEVs to meet the necessary flexibility*

requirements of the VRE system with the least system cost, in line with the current society, technology development, and policies.”

Review Comment 3:

(a) Both the supplementary material and the methods section lack a discussion of the core model, a derivation of the computational process, or an elaboration of the algorithm design. The methodology and its innovation need more justification. It is unclear how the authors applied and customized the Decision Optimization for Low Carbon Power and Hydrogen Nexus model, and reproducing the results is therefore difficult.

Author Response:

- We displayed the model framework in Figure 1 and discussed the core model, the flexibility charging model, the degradation model, the degradation cost calculation, and the computation of the flexibility value in the “Methods” Section, involving the objective function and operational constraints. We customized this model by considering EV flexible charging and battery degradation cost.
- The core model (i.e., DOLPHYN) is a sector-coupling planning and decision optimization model to minimize the cost of the low-carbon power network. By modeling the coupling and conversion of different energy sectors and revealing the competition and complementary among different technologies, under operational constraints, the most effective and efficient design is identified with least costs achieved. The optimization is realized by linear programming solved by Gurobi with barrier methods.
- We customized the DOLPHYN model to optimize the energy systems without flexible charging, with flexible charging but without charging constraint and degradation cost, with flexible charging and degradation cost separately. Corresponding least system costs (Y_0 , Y_{BEVND} , Y_{BEV} , Y_{FCEV} in Eq.1-4) are obtained by the DOLPHYN optimization model under relevant operational and policy constraints (Φ_0 , Φ_{BEV} , Φ_{FCEV} in Eq.1-4). Then the flexibility values are computed by the differentials of the least system costs (Eq. 5-7 in Section 5.2). To be more specific, the flexibility value without degradation is the difference value between the least costs of energy systems without flexible charging and with flexible charging but without degradation (Eq.5 in Section 5.2). The flexibility value considering degradation is the difference value between the least costs of energy systems without and with flexible charging (Eq.6 in Section 5.2).
- To make it clearer, we have added/revised the manuscript by
 - (1) Adding the text in the first paragraph of Section 2 to describe the DOLPHYN model: “*The DOLPHYN is a sector-coupling planning and decision optimization model to minimize the cost of the low-carbon power network (see Methods section). It optimized the costs to identify the most effective and efficient design and operation of the energy system, by modeling the coupling and conversion of different energy sectors and revealing the competition and complementary among different technologies. Linear programming solved by Gurobi with barrier methods is applied to address the optimization problem under four operational constraints (detailed constraints are available in Methods).*”.
 - (2) Adding the text in the first paragraph of Section 2 to describe the DOLPHYN model: “*Linear programming solved by Gurobi with barrier methods is applied to address the optimization problem under four operational constraints (detailed constraints are available in Methods).*”.

- (3) Adding a schematic diagram to show how the flexibility values are computed using the cross-scale framework as Supplemental Figure 1 in the revised Supplemental Information.

Supplemental Figure 1. Schematic for the flexibility value computation of FCEVs and BEVs with battery degradation considered. The DOLPHYN model is customized for energy systems with different flexible charging settings, i.e., energy system with flexible charging, without flexible charging, and with flexible charging and battery degradation cost considered. The degradation cost is computed by Eq.(8) in Section 5.3 with the cycle life L_{bat} obtained by a micro-level porous electrode theory-based model. After selecting several representative weeks from 7-year data of renewable generation and electricity demand using clustering techniques, the optimization problems under operational constraints are solved by Gurobi with battier methods. Then the least system costs of energy systems with and without flexible charging, and with flexible charging and degradation cost are obtained (Eq. 1-4 in Section 5.2). Finally, the flexibility values are figured out by computing the difference values between corresponding optimized least system costs (Eq. 5-7 in Section 5.2).

- (4) Providing an open-access version of the codes on the specific implementation of the proposed model and algorithm in the manuscript, with some exemplary illustrations.

(b) The unit cost calculation for battery degradation is also unclear,

Author Response: Battery degradation cost is computed by dividing the battery replacement cost by the cycle life (as shown in Eq. 8 in Section 5.3), which is obtained by the PET-based micro-level model. According to a marginal perspective, with the degradation cost when BEVs are charged within 60min as the benchmark, we compute the extra degradation costs associated with other charging durations over the benchmark.

(c) how did the result of Supplementary Table 6 come?

Author Response:

- Most of the base parameters in Supplementary Table 5 (i.e., Supplementary Table 6 of the original manuscript) are from the PETLION paper mentioned in Section 5.5. The rest are simply computed based on the base parameters. For instance, the thickness and porosity of the anode and cathode are from the PETLION paper mentioned in Section 5.5 (Ref. 70). Electrode length, width, and layers per cell are from the paper “Ciez RE., Steingart D., *Asymptotic Cost Analysis of Intercalation Lithium-Ion Systems for Multi-hour Duration Energy Storage, Joule, 4, 597-614 (2020)*”. For example, the volume is calculated by the product of electrode length, width, layers, and thickness. The active mass is calculated by the product of porosity, volume, and density. The capacity is the product of energy density and active mass. The cost is the product of the price and active mass.

Review Comment 4:

The interpretation of the figures is suggested to be clearer and more detailed, e.g., Fig 1a. The four scenarios are unknown in this figure, why the charging costs are four parallel straight lines, and what is the range of time dimensions considered?

Author Response:

- Figure 1 shows the scheme of the cross-scale framework for comparing the flexibility values of BEVs and FCEVs. The scenarios are not described in this figure and are provided in Table 1 in the revised manuscript. The framework contains a macro-level model and a micro-level model. At the macro level, the DOLPHYN model coupling energy production, storage, transmission, and demand-side sectors is applied. To involve the battery degradation cost, a micro level for battery degradation analysis is used to generate a part of inputs for the macro-level decision model, DOLPHYN. The charging cost are not parallel straight lines. The minimized system cost including CAPEX, OPEX, Emission, and degradation costs is the output of the DOLPHYN model. We compute the least system costs with and without EV flexible charging, and then obtain the cost reduction due to EV flexibility, which is noted as the flexibility value of EVs.
- For the time range, the system optimization model is implemented over a set of representative weeks at an hourly resolution selected based on *K*-means clustering from 7-year data of renewable generation and electricity demand, as mentioned in the first paragraph of Section 2 and the second paragraph of Section 5.1.
- To make it clearer, we have inserted the below text into the first paragraph of Section 2 to describe the representative week selection “*The representative weeks are selected based on K-means clustering from 7-year data of renewable generation and electricity demand.*”.

Review Comment 5:

How realistic are the assumptions stated in Section 2 and how do they lead to discrepancies when compared to "real world" events?

Author Response:

- This comment seems to be a criticism of this field rather than this paper specifically. It is not realistic to represent “real world” events since many factors including policies and people’s reaction are often unpredictable. However, there are reasonable boundaries in our study. For example, the deferrable EV demand was kept within a relatively small range in our study under

the assumption that the flexible charging demand is less than the total EV charging demand. Some main parameters of the power system are derived from the NREL annual technology baselines and the U.S. Energy Information Administration (EIA) Annual Energy Outlook 2018 for the year 2045. The hydrogen parameters are from IEA technical report 2019 and other papers listed in the Supplemental Information.

- Our assumptions are reasonable and well-founded. For instance, considering the high uncertainty of future EV charging distribution and the ratio of deferrable demand, our study on EVs flexibility is based on a marginal perspective. In addition, we discussed a wide range of scenarios, from the initial decarbonization process (in recent decades) with mixed hydrogen pathway, smaller deferrable demand, and lower carbon prices, to deep decarbonization process (in the long term) with electrolytic-only hydrogen pathway, larger deferrable demand, and higher carbon prices. Meanwhile, sensitivity analysis was also carried out for uncertain parameters, such as different battery replacement costs (Figure 3) and charging durations (Figure 2). Therefore, although we made some assumptions, some fluctuations of them might not affect our main conclusions.
- The aims of this work are to reveal how much battery degradation, which most studies often ignored, will affect the flexibility values of BEVs and the comparative advantages between BEVs and FCEVs, and to provide insights for priority selection between BEVs and FCEVs to meet the necessary flexibility requirements of the VRE system with the least system cost according to various decarbonization processes, which are already discussed in our scenarios.
- Last but not least, another contribution of this work is to highlight the significance of such a paradigm of our cross-scale framework on macroscopic analysis, which is not affected by our assumptions.

Review Comment 6:

In Section 3, the authors designed a large number of scenarios, but the introduction of the scenarios and the definition of key attributes are vague. For example, fast and normal charging, deferrable demand, charging duration, etc.

Author Response:

- Thank you for the comment. These scenarios are introduced before corresponding results are stated. We also summarize all these scenarios in Table 1 in the revised manuscript. The definition of fast and normal charging is described at the first paragraph of Section 2.1. Deferrable demand signifies the flexible EV consumption of hydrogen or electric power. Charging duration is the charging time for BEVs.
- To make it clearer, we have added/revised the manuscript by
 - (1) Adding the text at the end of the fifth paragraph of Section 2 to describe the case selection logic *“To model flexible EV charging, we considered two load-shifting settings at room temperature with the electrolytic-only hydrogen pathway applied. The charging duration of the fast-charging setting is less than 1 h, while that of the normal charging setting is 6 h). Different battery replacement costs from \$50/kWh to \$200/kWh are considered in the normal charging case. For the fast-charging case, an average charging scenario and an extreme fast-charging scenario are discussed. When comparing the flexibility values of BEVs and FCEVs under different hydrogen pathways or temperatures, the BEV average fast-charging setting with a \$100/kWh battery replacement cost is applied. Two hydrogen*

pathways include the electrolytic-only hydrogen and the mixed hydrogen pathway. Temperatures considered in this work are 25 °C, 10 °C, and 0 °C, respectively. Lower temperatures are discussed because studies have illustrated that lower temperatures evidently affect battery charging.”.

- (2) A Table (i.e., Table 1 in the revised manuscript) that summarizes all the scenarios for comparing flexibility values of BEVs and FCEVs was provided in the Section 2 of manuscript.
- (3) Adding the text at the end of the caption of Figure 1 to define “*Deferrable demand*” and “*LHV*” appeared in Figure 1 “*Deferrable demand here is the flexible consumption of hydrogen or electric power for EVs. Larger deferrable demand means a larger level of EV flexible charging. LHV is short for low heating value.*”.

REVIEWER COMMENTS

Reviewer #1 (Remarks to the Author):

Thank you, I am satisfied with the changes made

Reviewer #2 (Remarks to the Author):

My previous review highlighted three main concerns. The authors did a good job of addressing two of the three by improving the description of the methodology and the clarity of the presentation of the scenarios and results. The third concern on the treatment of FCEV flexibility value is not adequately addressed. The framing of the analysis as evaluating the marginal cost savings from added flexibility in a system with a set number of vehicles and stations alleviates my concern about the spatial resolution but I am not convinced by the explanation of how refueling flexibility in FCEVs is modeled.

Hydrogen refueling infrastructure costs are assumed to be fixed by number of FCEVs and not impacted by flexibility of refueling. When thinking about flexibility within hydrogen refueling there more than one step that can be shifted. The production of hydrogen from natural gas or electricity can happen at different times than the refueling. The sizing of storage at the refueling station will determine this flexibility making the refueling infrastructure cost tied to the flexibility. Some energy demands will be more tightly tied to exact refueling time such as the compressor energy use but these are smaller than the energy required for hydrogen production. This two-step process for FCEV refueling energy demands and how that interacts with the flexibility value of FCEV refueling needs a better explanation at the very least.

Potential refueling windows for FCEVs are not clear. In contrast to BEVs, FCEVs refuel during trips not while parked at a destination. The time windows where refueling can be deferred are therefore different between the two types of EVs. The paper reads like they are treated the same.

Extra comment that could improve the strength of the findings:

- Findings change with temperature. Adding context on what fraction of the year the region has the given ambient temperatures would provide more insight into the relevance of different findings for this case study. It is even better if the temperatures can be matched with the charging times.

Reviewer #3 (Remarks to the Author):

Thank you for revising the assignment. Following careful consideration, the reviewer suggests that publication in Nature Communications be discouraged.

The main concern with the work is the lack of clarity and coherence in the main arguments. While the author raises an interesting question about the flexibility comparison of two types of vehicles, the related statement, scenario settings, methodology, and conclusions require further refinement. Specifically, the definition of flexibility value based on delayed charging or refueling demands is debatable and does not necessarily capture the full picture of what constitutes FCEV flexibility. Furthermore, the choice of comparison samples seems to ignore important cost factors related to the hydrogen refueling process, which undermines the validity of your conclusions. Additionally, the conclusions appear to be too broad and do not account for the unique features of each vehicle type. The arrival of the vehicle, the generation of demand, the infrastructure, and the choice of refueling behavior are not mentioned or described in an oversimplified way. Finally, the results are not presented in a clear and informative way, making it difficult for readers to understand the main findings and trends.

Some detailed suggestions and concerns are listed as follows:

1. The work is largely a comparison scenario constructed on deferred demand to demonstrate the cost savings of flexible energy replenishment. However, the logic of scenario setting remains unclear. Is the BEV charging scenario intended for private vehicles, commercial vehicles, heavy-duty vehicles, or mixed flow (The case is based on trucks as I can see in the method description, but this does not appear to have been made clear in the earlier text). Is charging done during the day or at night? How is the availability of chargers defined, is it available at public or private charging stations? The charging requirements and demands of various settings vary and cannot be generalized.
2. The significance of the delayed hydrogen refueling time and the value-of-time of waiting have not been stated clearly in the case of FCEV, when the hydrogen refueling duration has been neglected. Is the loss of lifetime due to hydrogen quality or refueling pressure of FCEV quantified or ignored? Based on the proposed method, is it possible to draw an analogy between the flexibility value of diesel and hydrogen vehicles?
3. Regarding battery degradation, there is a certain relationship between the depth of discharge and battery aging. The deeper the discharge, the more intense the chemical reactions and ion movement inside the battery, leading to an increase in internal pressure and temperature, which may accelerate battery aging. Fast charging can indeed reduce the depth of discharge during the daytime by providing convenient energy replenishment, which may help reduce battery aging. However, the authors may have overlooked the specific advantages of fast charging due to the lack of consideration for the choice of charging stations. Instead, they tested different charging speeds in the same scenario. To fully evaluate the impact of fast charging on battery aging, it may be necessary to conduct more comprehensive tests.
4. Regarding the results, how was the horizontal coordinate range determined in Figures 2 and 3? In the lower-left subfigure of Figure 5, refer to the second column, why the cost reduction is not as significant as it is when the delayable demand is 5, 10, or 15 at the same temperature as when it is 0? In addition, the authors were advised to go beyond just listing the data in the figures and simple comparisons, the reasons behind the key findings were also expected.

Response to Reviewers for NCOMMS-23-03417A-Z

Response to Reviewer #1:

Review Comment: Thank you, I am satisfied with the changes made.

Author Response: Thank you for the comments and recommendation.

Response to Reviewer #2:

Review Comment 0: My previous review highlighted three main concerns. The authors did a good job of addressing two of the three by improving the description of the methodology and the clarity of the presentation of the scenarios and results. The third concern on the treatment of FCEV flexibility value is not adequately addressed. The framing of the analysis as evaluating the marginal cost savings from added flexibility in a system with a set number of vehicles and stations alleviates my concern about the spatial resolution but I am not convinced by the explanation of how refueling flexibility in FCEVs is modeled.

Author Response: We thank reviewer 2 for recognizing our work in addressing two other concerns. Regarding the treatment of FCEV flexibility value, we attached FCEV flexibility to the flexibility of the hydrogen supply chain including storage and transmission processes since the FCEV refueling is modeled as a dispatchable demand. The FCEV flexibility value includes the cost reduction benefits of the whole supply chain owing to FCEV flexible refueling. We have added detailed explanations in the revised manuscript with the following responses (Author Response to Review Comments 1&2).

Review Comment 1:

Hydrogen refueling infrastructure costs are assumed to be fixed by number of FCEVs and not impacted by flexibility of refueling. When thinking about flexibility within hydrogen refueling there more than one step that can be shifted. The production of hydrogen from natural gas or electricity can happen at different times than the refueling. The sizing of storage at the refueling station will determine this flexibility making the refueling infrastructure cost tied to the flexibility. Some energy demands will be more tightly tied to exact refueling time such as the compressor energy use but these are smaller than the energy required for hydrogen production. This two-step process for FCEV refueling energy demands and how that interacts with the flexibility value of FCEV refueling needs a better explanation at the very least.

Author Response:

- Thank you for your comment. Hydrogen storage and transmission are involved in the model as two main measures to address the spatiotemporal mismatch between hydrogen production and refueling. Regarding hydrogen storage, stationary storage

(including storage at the refueling station), gas or liquid trucks are considered. For hydrogen transmission, pipelines and trucks (including gas trucks and liquid trucks) are mainly considered, enabling hydrogen shifting in space and time while being shared across the whole hydrogen network to match hydrogen demand. In summary, four steps are incorporated in the H₂ sector of our model, i.e., production, storage, compression, and transmission.

- The H₂ demand for each zone is developed based on available fuel consumption data and hourly charging profiles for FCEVs and the relative penetration of FCEVs.
- We agree that the sizing of storage affects the flexibility value of FCEV. That is why the flexibility value of FCEVs is attached to the flexibility of the hydrogen supply chain in this work, while modeling FCEVs refueling as a dispatchable demand. The decision variables include the planning and scheduling variables of flexible hydrogen transmission and storage for minimizing the energy system cost. The sizing of hydrogen storage and the capacity of hydrogen transmission are involved as the variables to be optimized in the DOLPHYN model in response to the hydrogen flexible demand.
- We also displayed the variation of optimized hydrogen storage sizing and the power generation by VRE with increasing deferrable demand in different scenarios considering different hydrogen pathways in Figure 4. The optimized hydrogen storage size in Figure 4 (the fourth column) contains both stationary and mobile storage, covering all stored hydrogen at demand-side sectors and refueling stations. While mobile storage includes gas trucks (compressed H₂ trucks) and liquid trucks. As the flexible FCEV refueling demand increases, the hydrogen storage size decreases together with the reduced minimum system cost, indicating the interaction between the FCEV flexibility and the hydrogen supply chain.
- In summary, the flexibility value of FCEV flexible refueling is attached to the hydrogen supply chain. When deferrable refueling demand is applied, the total cost of the low-carbon energy system is reduced, with more flexibility provided by the FCEV fleet and possibly less flexibility requirement for other processes (on account of the relatively higher cost to achieve flexibility) in the hydrogen supply chain, reduced hydrogen storage in Fig.4 for instance. That is how the flexibility of FCEV deferrable refueling interacts with the hydrogen supply chain. To address this comment, we have revised the manuscript by:

(1) Inserting the below text into the sixth paragraph of the “Results-Model overview” section to specifically state the hydrogen storage and transmission scheduling: *“Specifically, hydrogen production, storage, compression, and transmission are included in the hydrogen supply chain, as shown in Figure 1 and supplementary Fig. 2. Among the four elements, hydrogen storage and transmission are two direct measures to provide the required flexibility for the supply chain. Regarding hydrogen storage, in addition to stationary storage, mobile storage including trucks and pipelines is also considered in our model, enabling hydrogen shifting in space and time while being shared across the whole hydrogen network to match hydrogen demand. In other words, trucks and pipelines are mainly modeled as both transmission and storage functions*

to provide flexibility to the hydrogen supply chain. The optimization model incorporates the hydrogen flexible storage and transmission scheduling. It means that the decision variables of the DOLPHYN model involve the capacities of hydrogen storage and transmission between zones. These variables are optimized by the model in response to the hydrogen flexible demand. The H₂ demand for each zone is developed based on available fuel consumption data and hourly charging profiles for mainly heavy-duty FCEVs and the relative penetration of FCEVs.”.

- (2) Adding a figure in the supplementary information (i.e., Supplementary Figure 2) to describe the hydrogen supply chain involved in the DOLPHYN model.

Supplementary Figure 2. The schematic of the hydrogen supply chain involved in the DOLPHYN model.

- (3) Inserting a subsection “Hydrogen supply chain and balance constraints” in the “Method” section to describe the hydrogen modeling in detail: “The model for hydrogen supply chain is involved as an essential part of the whole DOLPHYN model, incorporating all steps in the hydrogen supply chain including hydrogen production, compression, transmission, and storage, as shown in Supplementary Figure 2. Almost all critical technological options are considered in each step.

The total cost of hydrogen supply chain involves the cost of hydrogen generation, conversion, transmission, and storage. For hydrogen production, electrolyzer, SMR with and without CCS are modeled. Regarding hydrogen transmission, gas/liquid trucks and pipelines for flexible transmission are modeled.

The hydrogen supply chain is scheduled following the hydrogen balance constraint. For a specific zone at a moment, the amount of H₂ production $h_{z,t}^{GEN}$ plus the amount of transported H₂ (positive for imports) $h_{z,t}^{TRA}$ and the amount of H₂ discharged from storage $h_{z,t}^{DIS}$ should be equal to the amount of H₂ charged to storage $h_{z,t}^{CHA}$ plus the H₂ demand $D_{z,t}$ and minus the lost demand $h_{z,t}^{LOS}$.

$$h_{z,t}^{GEN} + h_{z,t}^{TRA} + h_{z,t}^{DIS} = h_{z,t}^{CHA} + D_{z,t} - h_{z,t}^{LOS} \quad (12) \text{ ”.}$$

- (4) Inserting the below text into the first paragraph of the “Results-Flexibility values under different hydrogen pathways” to further state the interaction between the FCEV flexibility value and the flexibility of the supply chain: *“For FCEVs, the optimized hydrogen storage (consisting of both stationary and mobile storage) is reduced as increasing of the deferrable demand. More reduced hydrogen storage is observed under scenarios with higher H₂ demands. This can be explained that with more flexibility provided by the FCEV fleet, less flexibility is required for the storage process owing to the relatively higher cost to achieve flexibility than FCEVs. The results illustrate how the FCEV flexibility value interacts with the hydrogen storage process and is attached to the flexibility of the supply chain.”*

Review Comment 2:

Potential refueling windows for FCEVs are not clear. In contrast to BEVs, FCEVs refuel during trips not while parked at a destination. The time windows where refueling can be deferred are therefore different between the two types of EVs. The paper reads like they are treated the same.

Author Response:

- From a marginal perspective, considering the whole EV fleet, always some vehicles in the fleet are going to be refueled or charged. Then the demands of FCEV refueling and BEV charging both exist every moment, either on the road or in parking lots.
- Like time-variant power price for BEV charging, the refueling deferral can be achieved by similar incentives, for instance, time- or space-variant hydrogen prices or coupons checked on related APPs. For vehicles not at the refueling station, the FCEV refueling can be scheduled through the interaction between an APP and the vehicle.
- The time windows where refueling/charging can be deferred for FCEVs and BEVs depend mainly on the willingness to defer of the drivers/operators, which should have no significant difference in nature. Therefore, the time window for FCEVs refueling and BEVs charging could be treated similarly.
- At last, even though the deferrable demand of FCEVs is different from that of EVs, the flexibility values of FCEVs can be also observed and compared in our results. For instance, we can compare the flexibility values of FCEVs and BEVs when the deferrable demand of FCEVs is 10 tonne H₂/hour and that of BEVs is 15 or 20 MW.
- To address this comment, we have revised the manuscript by:
 - (1) Inserting the below text into the first paragraph of “Method-Flexible charging modeling” section to specifically state the deferrable demand and time window: *“In other words, the deferrable demand is the unserved demand in the following time window.”*
 - (2) Inserting the below text after the last sentence of the first paragraph of “Method-Flexible charging modeling” section: *“Although the vehicle is not at the refueling station, the FCEV refueling can be scheduled through the interaction between the APPs and the vehicle. From the marginal perspective, always some vehicles in the EV fleet are going to be charged or refueled. Then*

the demands of FCEV refueling and BEV charging both exist every moment, either on the road or in parking lots. Therefore, the time window for FCEVs refueling and BEVs charging could be treated similarly despite the two mechanisms being different.”.

Review Comment 3:

Extra comment that could improve the strength of the findings:

- Findings change with temperature. Adding context on what fraction of the year the region has the given ambient temperatures would provide more insight into the relevance of different findings for this case study. It is even better if the temperatures can be matched with the charging times.

Author Response:

- Thank you for your comment. For our case study of U.S. Northeast region, the average temperature of New York overall a year is between 8-15 °C. Around a quarter of a year, the temperature in New York is around 25 °C. Around half of a year, the temperature in New York is about 10 °C. In winter, the temperature fluctuates around 0 °C. Therefore, when exploring the effects of service temperature on BEV flexibility values, three temperatures (i.e., 25 °C, 10 °C, and 0 °C) are used. The below text has been inserted into the first paragraph of the “Results-Effects of the service temperature on EV flexibility values” section to state the region temperature in this case study: *“For example, in our case study focused on the U.S. Northeast region, where the annual average temperature ranges between 8-15 °C, the local temperature hovers around 25°C for approximately a quarter of the year. For more than half the year, it remains near 10°C, and occasionally is below 0°C during winter.”.*
- Investigation on the EV flexibility values for the cases of temperatures matched with charging times is a good idea, which would be considered as future work. The below text has been inserted into the “Discussion” section of the revised manuscript: *“The effects of temperature and charging durations on BEV flexibility value are separate. How the coupling effects of the two factors are not involved in this work. A future work is to develop models and algorithms that integrate temperatures and charging times to evaluate EV flexibility values.”.*

Response to Reviewer #3:

Review Comment 0: Thank you for revising the assignment. Following careful consideration, the reviewer suggests that publication in Nature Communications be discouraged. The main concern with the work is the lack of clarity and coherence in the main arguments. While the author raises an interesting question about the flexibility comparison of two types of vehicles, the related statement, scenario settings, methodology, and conclusions require further refinement.

Author Response: Thank you for your comments and valuable suggestions. We have refined the arguments and statements in the revised manuscript. Following are point-to-point responses to your comments.

Review Comment 1:

(a) Specifically, the definition of flexibility value based on delayed charging or refueling demands is debatable and does not necessarily capture the full picture of what constitutes FCEV flexibility.

Author Response:

- Regarding the definition of flexibility value, flexibility is defined by International Energy Agency (IEA) as the ability to manage energy demand and generation according to local climate conditions, user needs, and grid requirements. Flexibility can reduce costs, raise revenues, and ensure the reliability of power systems [37]. Demand flexibility is the key to enabling a low-cost, low-carbon grid [R1]. Quantifying the value of EV flexibility as the energy system cost reduction owing to EVs' flexible charging or refueling (which is denoted by deferrable demand in this work) is reasonable, and similar approaches for demand flexibility valuation are widely adopted in literature [39,40,R2,R3]. The quantification of flexibility value is introduced in the first sentence of the first paragraph of the "Results-Model overview" section.

[37] Langevin, J. et al. US building energy efficiency and flexibility as an electric grid resource. *Joule* 5, 2102–2128 (2021).

[39] Gottwalt, S., Gärtner, J., Schmeck, H., et al. Modeling and valuation of residential demand flexibility for renewable energy integration. *IEEE Trans. Smart Grid*. 8(6), 2565-2574 (2017).

[40] Jenkins, J.D., Zhou, Z., Ponciroli, R., et al. The benefits of nuclear flexibility in power system operations with renewable energy. *Appl. Energy*, 222(15), 872-884 (2018).

[R1] Goldenberg, C., Dyson, M., Demand flexibility, RMI's new Insight Brief. 2018

[R2] Bolorinos, J., Mauter, M.. Integrated Energy Flexibility Management at Wastewater Treatment Facilities. *Environ. Sci. Technol. Online*, (2023)

[R3] Pavic, I., Capuder, T., Kuzle, I. Value of flexible electric vehicles in providing spinning reserve services. *Appl. Energy*. 157(1), 60-74 (2015)

- For the delayed charging or refueling, the delayed demand is the flexible hydrogen or electric power consumption for EVs in the following time window. Zero deferrable demand for reference and benchmark means no flexible EV charging or refueling applied.
- In response to the confusion of what constitutes FCEV flexibility, FCEVs flexibility in this work is attached to the flexibility of the hydrogen supply chain since FCEV refueling is modeled as a dispatchable demand, specifically the storage capacities and transmission between zones (As mentioned at the second sentence of the sixth paragraph of the “Results-Model overview” section). Four essential elements, i.e., hydrogen production, storage, compression, and transmission are incorporated in the modeling of the hydrogen supply chain. Two of which, hydrogen storage and transmission, are direct measures to provide the required flexibility for the hydrogen supply chain. Hydrogen trucks and pipelines play their roles not only in transmission but also in mobile storage for supplement stationary storage, offering flexibility for the supply chain. Such flexibility is preconditioned on the deployment of FCEVs fleet, and therefore, we attribute this flexibility to the FCEV pathway. Corresponding variables of the supply chain (hydrogen storage capacity, transported hydrogen amount, for instance) are involved in the decision variables to be optimized by the DOLPHYN model with the response to flexible refueling demand of FCEV fleet (part of the optimized variables are shown in Figure 4). When the FCEV flexible demand is increased, less flexibility is required in the supply chain owing to higher flexibility provided by FCEVs.
- To address this comment, we have revised the manuscript by
 - (1) Inserting the below text after the first sentence of the third paragraph of the “Introduction” section to clarify the rationality of the definition of flexibility value based on EV flexible demands: *“The valuation of demand flexibility is usually quantified by the cost reduction or benefits of the energy system owing to demand-side flexibility³⁹⁻⁴⁰.”*
 - (2) Inserting the below text after the second sentence in the sixth paragraph of the “Results-Model overview” section to specifically state the constitution of FCEV flexibility: *“Specifically, hydrogen production, storage, compression, and transmission are included in the hydrogen supply chain, as shown in Figure 1 and supplementary Fig. 2. Among the four elements, hydrogen storage and transmission are two direct measures to provide the required flexibility for the supply chain. Regarding hydrogen storage, in addition to stationary storage, mobile storage including trucks and pipelines is also considered in our model, enabling hydrogen shifting in space and time while being shared across the whole hydrogen network to match hydrogen demand. In other words, trucks and pipelines are mainly modeled as both transmission and storage functions to provide flexibility to the hydrogen supply chain. The optimization model incorporates the hydrogen flexible storage and transmission scheduling. It means that the decision variables of the DOLPHYN model involve the capacities of hydrogen storage and transmission between zones. These*

variables are optimized by the model in response to the hydrogen flexible demand. The H₂ demand for each zone is developed based on available fuel consumption data and hourly charging profiles for mainly heavy-duty FCEVs and the relative penetration of FCEVs.”.

- (3) Inserting the below text into the third paragraph of the “Results-Model overview” section to specifically state the deferrable demand: *“In other words, the deferrable demand is the unserved demand in the following time window.”.*

(b) Furthermore, the choice of comparison samples seems to ignore important cost factors related to the hydrogen refueling process, which undermines the validity of your conclusions.

Author Response:

- We involve general costs related to the hydrogen refueling process from hydrogen production and storage to hydrogen transmission. The modeling of the hydrogen supply chain incorporates the whole hydrogen stream, i.e., hydrogen production, storage, compression, transmission, and demand-side use, as shown in Figure 1. While among these elements, hydrogen storage and transmission are closely related to FCEV flexible refueling.
- As for the investment in hydrogen refueling infrastructures, the system cost we optimized is the operational cost without investment in infrastructures, under the assumption that the FCEV refueling infrastructures are already available. Besides, the costs of charging or refueling infrastructures are fixed values (as shown in supplementary Table 1-3) depending on the total BEV charging demand or FCEV refueling demand, rather than the flexible part of demands. From a marginal perspective we used in this study, flexibly operating the refueling infrastructure does not directly add to the refueling infrastructure capacity or cost. Since the cost of hydrogen refueling infrastructure is fixed and unsusceptible to the flexible scheduling of FCEVs, it is not a significant factor (or cost) when assessing the flexibility values of FCEVs.
- To sum up, the important cost factors related to FCEV flexible refueling are already involved in our model and analysis. Corresponding decision variables (generation capacity, storage capacity, transported hydrogen amount, for instance) are all optimized by the proposed model, under a series of constraints, with part of the results shown in Figure 4.
- To address this comment, we have revised the manuscript by:
 - (1) Inserting the below text into the sixth paragraph of the “Results-Model overview” section to specifically state the cost factors involved in the model: *“To be specific, hydrogen production, storage, compression, and transmission are included in the hydrogen supply chain, as shown in Figure 1.”.*
 - (2) Adding a figure in the supplementary information (i.e., Supplementary Figure 2) to describe the hydrogen supply chain involved in the DOLPHYN model.

Supplementary Figure 2. The schematic of the hydrogen supply chain involved in the DOLPHYN model.

- (3) Inserting a subsection “Hydrogen supply chain and balance constraints” in the “Method” section to describe the cost factors, hydrogen modeling and constraints in detail: *“The model for hydrogen supply chain is involved as an essential part of the whole DOLPHYN model, incorporating all steps in the hydrogen supply chain including hydrogen production, compression, transmission, and storage, as shown in Supplementary Figure 2. Almost all critical technological options are considered in each step.*

The total cost of hydrogen supply chain involves the cost of hydrogen generation, conversion, transmission, and storage. For hydrogen production, electrolyzer, SMR with and without CCS are all modeled. Regarding hydrogen transmission, gas/liquid trucks and pipelines for flexible transmission are modeled.”.

(c) Additionally, the conclusions appear to be too broad and do not account for the unique features of each vehicle type. The arrival of the vehicle, the generation of demand, the infrastructure, and the choice of refueling behavior are not mentioned or described in an oversimplified way.

Author Response:

- The conclusions have been largely expanded and revised with further explanations including unique features of FCEVs and BEVs. How policies and technologies influence the comparative advantages of BEVs and FCEVs in terms of their flexibility values are summarized. For FCEVs, the coupling between hydrogen and power sectors is thoroughly discussed. The battery degradation of BEVs under different scenarios is also discussed in a detailed way. The “Discussion” section (i.e. the Conclusion section in the original manuscript) has been largely revised with detailed points listed following in the Author Response to Review Comment 1(d).
- From the marginal perspective, always some vehicles in the EV fleet arrive and are going to be charged or refueled. We use the concept of deferrable demand for flexible charging or refueling demand in a specific time window. Then the arrival of vehicles is not the key point. Only for the “BEV Average” scenario, we assume BEV arrivals

are uniformly distributed within an hour, so that we can average the extra degradation cost of 15-min, 30-min, 45-min and 60-min charging duration.

- For the generation of demand, the electricity demand data are based on 2018 NREL electrification futures study load projection for 2050 [Ref. 65 in the manuscript]. While the H₂ demands are developed based on available fuel consumption data, hourly refueling profiles for FCEVs, and the relative penetration of FCEVs (Supplementary Figure 3).
- For the infrastructure, this work is preconditioned on the existence of BEV and FCEV fleets and charging/refueling infrastructure in the future. Then the investments of infrastructures are fixed values (as shown in supplementary Table 1-3) depending on total BEV charging demand or FCEVs refueling demand, rather than the flexible part of demands. The least cost we optimized does not include additional investments in infrastructures. Therefore, this is not an important factor in our marginal analysis of the flexibility values of EVs owing to their flexible charging or refueling.
- Always some FCEVs are going to get refueled, from the marginal analysis. The choice of refueling or not can be scheduled by the incentive signals from APPs.
- To address this comment, we have /revised the manuscript by
 - (1) Inserting the below text after the last sentence in the first paragraph of the “Method-Flexible charging modeling” section to further explain the arrival of vehicles and the refueling or charging demand: *“From the marginal perspective, always some vehicles in the EV fleet are going to be charged or refueled. Then the demands of FCEV refueling and BEV charging both exist every moment, either on the road or in parking lots. Therefore, the time window for FCEVs refueling and BEVs charging could be the same despite the two mechanisms being different.”*.
 - (2) Inserting the below text at the last paragraph in the “Method-Flexible charging modeling” section to state the generation of demand: *“The electricity demand data are based on 2018 NREL electrification futures study load projection for 2050⁶⁵. While the H₂ demands are developed based on available fuel consumption data, hourly refueling profiles, and the relative penetration of FCEVs⁶⁶ (Supplementary Figure 3).”*.
 - (3) Adding a figure in the supplementary information (i.e., Supplementary Figure 3) to show the hourly H₂ demands at different zones.

Supplementary Figure 3. Hydrogen demand at different zones. Hourly H₂ demands profiles for each zone. Corresponding zones are shown in Ref. 50.

(d) Finally, the results are not presented in a clear and informative way, making it difficult for readers to understand the main findings and trends.

Author Response:

- Thank you for pointing this out. We have rewritten the results to present them in a clearer and more informative way and have largely expanded/revised the Discussion section (i.e., the Conclusion section in the original manuscript) with further explanations of these findings. The detailed revisions in the “Results” section are listed in the Author Response to Review Comment 5.
- In addition, we have also revised the manuscript in the “Discussion” section by:
 - (1) Inserting the below text in the third sentence in the first paragraph of the Discussion section to clarify involved variables: “*after optimizing decision variables related to power and hydrogen generation, storage capacity, and hydrogen transmission.*”.
 - (2) Rewriting the effects of BEV fast charging, charging temperatures, and battery replacement cost on relative advantages of BEVs and FCEVs as the flexibility provider in the third paragraph of the Discussion section as “*Fast charging and lower charging temperatures significantly diminish the flexibility value of BEVs. As a result, under scenarios with extremely short charging durations and low temperatures, FCEVs are more promising flexibility providers, which is primarily due to the substantially reduced battery lifespan in such conditions. Furthermore, escalating battery replacement costs can also reduce BEV flexibility value. Typically, battery replacement costs remain relatively consistent over time. To ensure a prolonged battery lifespan and greater BEV flexibility, it’s advisable to avoid conditions of low temperatures and fast charging. However, this is not always feasible. Thus, advancements in battery technology, particularly concerning charging and thermal performance, are important to bolster BEVs’ relative advantages as flexibility provider, since the*

lifespan and associated degradation costs of BEV batteries is significant when assessing the flexibility value.”

- (3) Revising the discussions of BEV and FCEV flexibility values under various policies (including H₂ demand and carbon price) with detailed explanation in the third paragraph of the Discussion section as *“When the H₂ demand is high (2 or 4 Mtonne/year) and the carbon price is low (\$100/tonne or \$0/tonne), FCEVs show more potential to providing essential flexibility. In most other scenarios, BEVs tend to have a superior flexibility value. Higher carbon prices, which are often associated with medium to deep decarbonization levels, promote a larger penetration of VRE, increased battery storage capacity, and reduced reliance on CCGT for power generation. This surge in flexibility requirements then favors BEVs. Higher H₂ demands lead to tighter integration between the hydrogen and power sectors. Consequently, in such scenarios, the flexibility value of BEVs significantly decreases, making them less competitive compared to FCEVs.”*
- (4) Inserting the below text into the third paragraph of the Discussion section for further explanation of the effects of H₂ pathway: *“Regarding the H₂ generation pathway, when the mixed hydrogen pathway is replaced by the electrolytic hydrogen-only pathway, the flexibility value of BEVs is more evidently reduced, on account of the stronger coupling of the hydrogen supply chain to the power system through electrolysis due to higher hydrogen storage and less battery storage.”*
- (5) Deleting the below text in the fourth paragraph of the Discussion section: *“First, the preference for BEVs or FCEVs depends on the hydrogen pathway and carbon price, which reflects the progress in decarbonization. At initial or mid-term stages with lower deferrable demand, moderate carbon prices, and mixed hydrogen pathways, BEVs could provide higher flexibility values. However, for deeper decarbonization with the electrolytic hydrogen-only pathway and increased deferrable demand, FCEVs show more potential in providing flexibility under relatively more scenarios.”*
- (6) Deleting the below text in the fifth paragraph of the Discussion section: *“Relevant technological development, such as intelligent battery fast charging and thermal management, is also of great importance for determining which kind of transportation electrification could provide higher flexibility values, since the lifetime and corresponding degradation cost of BEV batteries is significant when assessing the flexibility values.”*

Review Comment 2:

The work is largely a comparison scenario constructed on deferred demand to demonstrate the cost savings of flexible energy replenishment. However, the logic of scenario setting remains unclear. Is the BEV charging scenario intended for private vehicles, commercial vehicles, heavy-duty vehicles, or mixed flow (The case is based on trucks as I can see in the method description, but this does not appear to have been

made clear in the earlier text). Is charging done during the day or at night? How is the availability of chargers defined, is it available at public or private charging stations? The charging requirements and demands of various settings vary and cannot be generalized.

Author Response:

- The logic of scenario setting is based on variable control. The scenarios are summarized in Table 1. As described in the “Results-Scenario setup” section, we selected five kinds of scenarios to compare the flexibility values of BEVs and FCEVs under various charging modes, battery replacement costs, hydrogen pathways, and service temperatures. Two charging modes were evaluated, i.e., fast charging and normal charging. For fast charging, the charging load may be shifted to any time within one-hour window starting from vehicle arrival. Two cases namely “BEV Average” and “BEV Extreme” are involved in the fast-charging setting. While for the normal-charging scenario, different battery replacement costs are considered with the time window being 6 hours. Mixed and electrolytic-only hydrogen pathways are compared for different hydrogen-pathway scenarios with other scenario variates fixed. Finally, under the electrolytic-only pathway, the impacts of different BEV service temperatures on the flexibility value of BEVs are evaluated.
- The presented analyses mainly consider heavy-duty EVs. Different charging scenarios are designed for short- and long-haul heavy-duty EVs. For instance, a six-hour time window is more suitable for short-haul vehicles [Ref. 63 in the manuscript], and a one-hour time window is mainly for long-haul ones.
- The charging can be done either during the day or at night, according to the demand and the incentives of power or hydrogen prices, as shown in the left panel of Figure 1b. The H₂ demands are developed based on available fuel consumption data, hourly refueling profiles, and the relative penetration of FCEVs [Ref. 66, Supplementary Figure 3]. The electricity demand data are based on the 2018 NREL electrification futures study load projection for 2050 [Ref. 65 in the manuscript].
- Our work is on the premise that the chargers are enough and available, either at public or private charging stations. From the marginal perspective, the chargers are not a bottleneck, either, since the EV flexible charging is modeled as the deferrable demand. It means that BEVs do not need to be charged immediately after their arrival, as stated in the first sentence of the “Method-Flexible charging modeling” section.
- Precisely because charging requirements and demands vary under different settings, we classified different scenarios for flexibility value analysis. For instance, for different types of EVs with corresponding charging duration requirements, we categorized EV charging into fast charging and normal charging.
- To address this comment, we have revised the manuscript by:
 - (1) Inserting “*mainly for heavy-duty EVs*” to clarify the target EV type in the first sentence of the “Results-Scenario setup” section.
 - (2) Inserting the below text after the first sentence of the “Results-Scenario setup” section to indicate the logic of the scenario setting: “*In view of different charging requirements and demands, we classified five scenarios for flexibility value evaluation based on variable control.*”.

- (3) Inserting the below text in the last paragraph in the “Method-Flexible charging modeling” section to state the generation of demand: “*The electricity demand data are based on 2018 NREL electrification futures study load projection for 2050⁶⁵. While the H₂ demands are developed based on available fuel consumption data, hourly refueling profiles, and the relative penetration of FCEVs⁶⁶(Supplementary Figure 3).*”.

Review Comment 3:

The significance of the delayed hydrogen refueling time and the value-of-time of waiting have not been stated clearly in the case of FCEV, when the hydrogen refueling duration has been neglected. Is the loss of lifetime due to hydrogen quality or refueling pressure of FCEV quantified or ignored? Based on the proposed method, is it possible to draw an analogy between the flexibility value of diesel and hydrogen vehicles?

Author Response:

- The delayed hydrogen refueling time and the value-of-time of waiting mean the longest time FCEVs are refueled (1h or 6h in this work) through dispatch. Delayed FCEV refueling denotes FCEV flexible refueling. Thus, the significance of the delayed hydrogen refueling and the value-of-time of waiting is indeed quantified as the flexible values of FCEVs due to FCEV flexible refueling in this work.
- The loss of lifetime due to FCEV flexible refueling is ignored in our work since flexible refueling does not necessarily have impact on the refueling speed or pressure. The demand deferral shifts the FCEV refueling time only; it does not shorten or extend it. The hydrogen quality and refueling pressure of FCEV are fixed and unaffected by flexible refueling. Thus, FCEV flexible refueling is assumed to have not impact on its lifetime.
- Gasoline vehicles are similar to FCEVs in the deferrable refueling definition and neglect of aging compared with BEVs. Therefore, based on the proposed method, it is possible to draw an analogy between the flexibility values of diesel and hydrogen vehicles. However, the flexibility values of gasoline vehicles to the energy systems are smaller than those of FCEVs, since gasoline is less coupled with renewable power generation and the future low-carbon power systems compared to hydrogen.
- To address this comment, we have revised the manuscript by inserting the below text after the last paragraph of the “Methods-Degradation cost calculation” section to introduce the lifetime setting in the hydrogen supply chain: “*Extra degradation of FCEV flexible charging was not considered since the hydrogen quality and refueling pressure of FCEVs are fixed and unaffected by these variables. The demand deferral shifts the FCEV refueling time only; it does not shorten or extend it. The lifetimes of electrolyzers and fuel cells are simplified as a fixed parameter (e.g., the lifetime of the fuel cell is approximated as 10 years), as shown in Supplementary Table 2.*”.

Review Comment 4:

Regarding battery degradation, there is a certain relationship between the depth of discharge and battery aging. The deeper the discharge, the more intense the chemical reactions and ion movement inside the battery, leading to an increase in internal

pressure and temperature, which may accelerate battery aging. Fast charging can indeed reduce the depth of discharge during the daytime by providing convenient energy replenishment, which may help reduce battery aging. However, the authors may have overlooked the specific advantages of fast charging due to the lack of consideration for the choice of charging stations. Instead, they tested different charging speeds in the same scenario. To fully evaluate the impact of fast charging on battery aging, it may be necessary to conduct more comprehensive tests.

Author Response:

- Thank you for your comment. In this work, we analyzed the flexibility values of BEVs preconditioned on the assumption of enough chargers and charge stations. The charger protocols are simplified strategies defined by various charging duration from 10 minutes to 6 hours. All the BEVs are simulated to be charged from 30% state of charge (SOC) to 80% SOC using different charging protocols under various charging time constraints. Charging from 30% SOC to 80% SOC is the most common fast-charging setting in battery tests and is suggested for practical vehicle usage (Duru Kamala Kumari, et al, IEEE Trans. Device Mater. Reliab., 21, 137-152, 2021). We have inserted the below text before the last sentence in the fourth paragraph of the “Method-Porous electrode theory-based battery degradation model” section to introduce the depth of discharging: *“The cells are simulated to be charged from 30% state of charge (SOC) to 80% SOC using different charging protocols under various time constraints.”*.
- We agree that fast charging can reduce the depth of discharge by providing convenient energy replenishment, which may help reduce battery aging. However, it is not the main concern in this work. Our consideration for EV charging or not is mainly according to the macro-control of the power price of the low-carbon energy system. In future work, we would consider the factor of depth of battery discharge by providing convenient energy replenishment from fast charging or charging station choice. We have inserted the below text at the end of the “Discussion” section of the revised manuscript: *“In addition, investigation on the depth of battery charging and discharging and its influence on BEV flexibility is considered as another future work.”*

Review Comment 5:

Regarding the results, how was the horizontal coordinate range determined In Figures 2 and 3? In the lower-left subfigure of Figure 5, refer to the second column, why the cost reduction is not as significant as it is when the delayable demand is 5, 10, or 15 at the same temperature as when it is 0? In addition, the authors were advised to go beyond just listing the data in the figures and simple comparisons, the reasons behind the key findings were also expected.

Author Response:

- Thank you for your comment. The horizontal coordinate in Figure 2 and Figure 3 is the deferrable demand, which signifies the flexible consumption of hydrogen or electric power for EVs. Thus, the range of deferrable demand is determined by the possible level of EV flexible hydrogen or power consumption from a marginal perspective. If the deferrable demand is too large, still using marginal analysis might

encounter some problems due to nonlinear property. As mentioned in the third paragraph of “Results-Model overview” section, the scale of deferrable EV demand was kept within a relatively small range in our analysis.

- Figure 5 is equivalent to Supplementary Figure 4 in the supplementary information (SI). It is evidently observed that when the carbon price is 0 and the hydrogen demand is 4 Mtonne/year, the benefits of BEVs flexible charging on system cost reduction is small enough to be ignored. That is because, in these scenarios with higher hydrogen demand and relatively lower carbon price, less share of VRE is encouraged in the energy system due to cost-effectiveness, with relatively less flexibility required. Meanwhile, with higher H₂ demand, the flexibility provided by FCEVs might be enough for the energy system. Thus, FCEVs play a more important role in providing the required flexibility for the lower carbon energy system. Then the flexibility of the system is pretty enough. So, the cost reduction of BEVs is less significant compared with the second column.
- Thank you for your valuable advice on this paper to add reasons behind these findings. We have replenished the explanation of the results in the revised version as follows:
 - (1) Inserting the below text before the third sentence in the fourth paragraph of the “Methods-Flexible charging modeling” section to further explain the reason why the scale of EV deferrable demand is within a relatively small range: “*The maximum deferrable demand is a fraction of the available capacity in a particular time step. To avoid problems due to nonlinear properties during marginal analysis, the scale of deferrable EV demand is kept within a relatively small range in our analysis.*”.
 - (2) Inserting the below text after the second sentence in the fourth paragraph of the “Results-Flexibility values of BEVs and FCEVs under fast and normal charging settings” section for further explanation: “*Relatively less flexibility is required when the carbon price is lower, owing to VRE discouraging. Under this circumstance, a higher hydrogen demand implying the strong coupling of power and hydrogen sectors makes FCEV predominate over BEVs as the flexibility provider.*”.
 - (3) Inserting the below text at the end of the first paragraph of the “Results-Flexibility values under different hydrogen pathways” section for further explanation: “*For FCEVs, the optimized hydrogen storage (consisting of both stationary and mobile storage) is reduced as increasing of the deferrable demand. More reduced hydrogen storage is observed under scenarios with higher H₂ demands. This can be explained that with more flexibility provided by the FCEV fleet, less flexibility is required for the storage process owing to the relatively higher cost to achieve flexibility than FCEVs. The results illustrate how the FCEV flexibility value interacts with the hydrogen storage process and is attached to the flexibility of the supply chain.*”.
 - (4) Inserting the below text after the third sentence in the second paragraph of the “Results-Effects of the service temperature on EV flexibility values” section for further explanation: “*As temperatures decrease, the accelerated battery*

degradation diminishes the benefits of BEVs as flexibility providers. The disparity in BEV flexibility values between 10°C and 0°C is more pronounced than that between 25°C and 10°C, due to the nonlinear increase in degradation costs as temperatures drop.”.

- (5) Inserting the below text after the last sentence in the second paragraph of the “Results- Effects of the service temperature on EV flexibility values” section for further explanation: *“The clear reduction in BEV flexibility values with decreasing charging temperatures indicates that for practical applications aiming to enhance BEV flexibility, effective thermal management should be adopted to prevent BEV batteries from operating and charging at low temperatures.”.*

REVIEWERS' COMMENTS

Reviewer #3 (Remarks to the Author):

Thank you for the revisions, I am satisfied with the revised version.

Response to Reviewers for NCOMMS-23-03417B

Response to Reviewer #3:

Review Comment: Thank you for the revisions, I am satisfied with the revised version.

Author Response: Thank you for your comments that had greatly helped improve the quality of our manuscript.